# Sequential Covariate Shift Detection Using Classifier Two-Sample Tests

## Abstract

A standard assumption in supervised learning is that the training data and test data are from the same distribution. However, this assumption often fails to hold in practice, which can cause the learned model to perform poorly. We consider the problem of detecting *covariate shift*, where the covariate distribution shifts but the conditional distribution of labels given covariates remains the same. This problem can naturally be solved using a two-sample test—*i.e.*, test whether the current test distribution of covariates equals the training distribution of covariates. Our algorithm builds on *classifier tests*, which train a discriminator to distinguish train and test covariates, and then use the accuracy of this discriminator as a test statistic. A key challenge is that classifier tests assume given a fixed set of test covariates. In practice, test covariates often arrive sequentially over time—e.g., a self-driving car observes a stream of images while driving. Furthermore, covariate shift can occur multiple times—*i.e.*, shift and then shift back later or gradually shift over time. To address these challenges, our algorithm trains the discriminator online. Furthermore, it evaluates test accuracy using each new covariate before taking a gradient step; this strategy avoids constructing a held-out test set, which can reduce sample efficiency. We prove that this optimization preserves the correctness—*i.e.*, our algorithm achieves a desired bound on the false positive rate. In our experiments, we show that our algorithm efficiently detects covariate shifts on ImageNet.

## 1 Introduction

A key challenge facing deep neural networks is their sensitivity to changes in the data distribution. In particular, supervised learning traditionally assumes that the training and test data are from the same distribution (Vapnik, 1998), but this assumption often fails in practice. For example, an autonomous car using perception to identify obstacles needs to be robust to shifts such as changes in the weather and lighting conditions. We focus on *covariate shift* (Shimodaira, 2000), where there is a shift in the covariate distribution $p(x)$, and the conditional label distribution $p(y \mid x)$ remains unchanged. Covariate shift can reduce model performance (Sugiyama & Müller, 2005), invalidate uncertainty estimates (Ovadia et al., 2019; Park et al., 2020), and affect model selection (Sugiyama et al., 2007).

One strategy is to devise an algorithm to detect covariate shift; if detected, the algorithm can alert the user that predictions may be unreliable. Covariate shift detection can be formulated as two-sample hypothesis test (Gretton et al., 2012a; Rabanser et al., 2018; Liu et al., 2020), where the goal is to determine whether two sets of examples are from the same distribution. To test for covariate shift, we choose the first sample to be the data used to train the model and the second sample to be recent test data given as input to the model. Then, the detector returns "covariate shift" if the hypothesis test indicates that the two samples are from different distributions and "no shift" otherwise.

We propose a detection algorithm based on classifier tests (Lopez-Paz & Oquab, 2017; Cheng & Cloninger, 2019; Kim et al., 2021), which use the accuracy of a classifier trained to distinguish the two samples as the test statistic. In particular, if the two samples are from the same distribution, then the accuracy should be $1/2$; otherwise, it should be $> 1/2$. Since the test statistic follows a binomial distribution, we use the Clopper-Pearson interval (Clopper & Pearson, 1934) (an exact confidence interval for the unknown success probability of the Binomial distribution) to derive the cutoff. In contrast, prior work relies on asymptotics to derive the cutoff, which results in approximations.

A key challenge is that the test examples are typically obtained over time—e.g., an autonomous robot continuously perceives its environment, and we want to detect if its distribution of observations shifts at any time. There are two key challenges to leveraging classifier tests in this setting. First, they rely on training a classifier to distinguish training and test examples; doing so on every step would be computationally intractable. Second, they rely on a held-out test set to estimate the test statistic, but constructing such a set online would reduce sample efficiency.

Rather than train a classifier at each step, our proposed algorithm trains a model online using stochastic gradient descent. Then, rather than construct a held-out test set, our algorithm evaluates the accuracy of the model online using each example before taking a gradient step on that example. We prove that this strategy results in an unbiased estimate of the model accuracy; as a consequence, the finite-sample guarantees on the false positive rate provided by the sequential test continue to hold. In addition, we prove bounds on the false negative rate under mild conditions on the classifier (i.e., it achieves nontrivial accuracy distinguishing the two distributions).

We evaluate our approach on both synthetic and natural shifts on the ImageNet (Russakovsky et al., 2015) dataset. In particular, we demonstrate that our approach achieves better sample efficiency than baseline algorithms; furthermore, it uniformly satisfies the desired false positive rate. Thus, our algorithm is an effective strategy for sequential covariate shift detection.

**Contributions.** We formulate (sequential) covariate shift detection as a two-sample test, and propose a novel algorithm to solve this problem (Section 3). Then, we prove finite sample bounds on false positive rate and false negative rate achieved by our algorithm (Section 4). Finally, we empirically demonstrate that our algorithm effectively detects shifts on ImageNet (Section 5).

**Sequential detection vs. sequential tests.** While we consider the sequential setting, we deliberately choose not use a sequential hypothesis test, since the covariate shift may occur after a delay or gradually over time. A sequential test only applies if *all* of the test data is shifted. Furthermore, since we are not using sequential tests, the false positive rate bound only holds per-step, not uniformly across all steps. This is necessary: we cannot guarantee that we detect a covariate shift occurring at a later point in time if we constrain the false positive to be bounded uniformly across all steps. In our experiments, we show that the rate of false alarms remains manageable while enabling our algorithm to detect covariate shift in a number of interesting scenarios.

## 2   RELATED WORK

**Covariate shift.** There has been work on training models in the presence of covariate shift. In particular, in the unsupervised domain adaptation setting (Ben-David et al., 2007; Bickel et al., 2007; Ganin et al., 2016), the algorithm has access to labeled examples from the source domain but only unlabeled examples from the target domain, and the goal is to train a model that achieves good performance on the target domain. One strategy is to use importance weighting to upweight source examples that are more similar to target examples (Bickel et al., 2007). Another strategy is to first learn an invariant representation (Ganin et al., 2016), which is an embedding space where the source and target examples are similar, and then train a model on this embedding space using the source examples. If we detect covariate shift, one solution is to retrain the model using these techniques.

**Two-sample tests.** We focus on *classifier two-sample tests (C2ST)*. In this approach, the idea is to train a binary classifier to distinguish source and target samples, compute a real-valued score based on this classifier as the test statistic, and then use a univariate two-sample test to determine the cutoff for rejecting the null hypothesis (Friedman, 2004). A natural test statistic is the classifier's accuracy on a held-out test set (Kim et al., 2021; Lopez-Paz & Oquab, 2017), or the differences in the classifier's logits (Cheng & Cloninger, 2019); in this work, we use the former. One way to compute the cutoff is to use the asymptotic distribution of the test statistic (Lopez-Paz & Oquab, 2017). Nonparametric tests such as permutation tests can also be used (Kim et al., 2021).

Another kind of two-sample test is called a *kernel two-sample test*. In this approach, the idea is to use the maximum mean discrepancy (MMD) between the two samples according to a given kernel embedding as the test statistic (Gretton et al., 2012a; Chwialkowski et al., 2015; Jitkrittum et al., 2016). The key design decision is the choice of kernel. One strategy is to use a nonparametric kernel such as Gaussian radial basis functions (Gretton et al., 2012a); alternatively, the kernel can also be optimized to minimize the false negative rate of the resulting test (Gretton et al., 2012b). In addition,

recent work has shown how to first learn a kernel function in the form of a deep neural network, and then evaluate the MMD distance on a held-out test set (Liu et al., 2020). The test statistic can be chosen based on finite sample bounds or based on its asymptotic distribution (Gretton et al., 2012a). Nonparametric permutation tests can also be used (Liu et al., 2020) with this approach.

**Concept Drift.** In the context of *concept drift* (Gama et al., 2014), there has been work detecting shifts in $p(x, y)$ (Gonçalves Jr et al., 2014). However, this work assumes that ground truth labels are provided for test examples, whereas our approach only requires unlabeled test examples. The former is substantially easier, since it suffices to check for drift in the distribution of prediction errors, which is usually very simple (e.g., a Bernoulli distribution for the 0-1 loss), making it easy to test for drift. In contrast, our approach checks for drift in high-dimensional covariates distribution.

**Sequential hypothesis testing.** A closely related problem is sequential hypothesis testing, which adaptively decides whether to reject the null hypothesis as samples become available (Wald, 1945). These approaches can also applied to two-sample testing (Balsubramani & Ramdas, 2015; Lhéritier & Cazals, 2018; 2019; Manole & Ramdas, 2021). However, as discussed above, they assume that the each distribution of the two samples does not change over time. In contrast, we are interested in the setting where the test examples might initially be from the same distribution as the training examples, but then shift at a later point in time. Sequential tests are not applicable to this setting.

## 3 SEQUENTIAL COVARIATE SHIFT DETECTION

### 3.1 PROBLEM FORMULATION

Let $\mathcal{X}$ be the covariate space, $\mathcal{S}$ be the source distribution over $\mathcal{X}$, and $\mathcal{T}_{t_1:t_2} = (\mathcal{T}_{t_1}, \mathcal{T}_{t_1+1}, \ldots, \mathcal{T}_{t_2})$ be a sequence of target distributions over $\mathcal{X}$ from time steps $t_1$ to $t_2$. On time step $t$, we consider samples $x_t \sim \mathcal{S}$ and $x'_t \sim \mathcal{T}_t$; in practice, $\mathcal{S}$ can be taken to be the uniform distribution over the training set. We let $S_{w,t} = (x_{t-w+1}, x_{t-w+2}, \ldots, x_t)$ and $T_{w,t} = (x'_{t-w+1}, x'_{t-w+2}, \ldots, x'_t)$ denote the recent examples in a time window of a given size $w \in \mathbb{N}$.

Our goal is to detect covariate shift at any step $t$. More precisely, we want to determine whether $\mathcal{S} \neq \bar{\mathcal{T}}_{w,t}$, where $\bar{\mathcal{T}}_{w,t} = \sum_{k=t-w+1}^{t} \mathcal{T}_k/w$, *i.e.*, whether the average target distributions over the previous $w$ steps is shifted compared to $\mathcal{S}$. For a fixed step $t$, this problem is a two-sample hypothesis test (Lehmann & Romano, 2006), where the null hypothesis is $H_0 : \mathcal{S} = \bar{\mathcal{T}}_{w,t}$, and the alternative hypothesis is $H_1 : \mathcal{S} \neq \bar{\mathcal{T}}_{w,t}$. That is, a two-sample test $\hat{f}$ is designed to compute

$$\hat{f}(S_{w,t}, T_{w,t}) \approx \begin{cases} 1 & \text{if } \mathcal{S} \neq \bar{\mathcal{T}}_{w,t} \\ 0 & \text{otherwise.} \end{cases}$$

Our goal is to design a two-sample test $\hat{f}$ for detecting covariate shift with this data stream. While we can in principle use any two-sample test, our goal is to design one that is both sample and computationally efficient while achieving high accuracy for high-dimensional data such as images. In addition, we want the test $\hat{f}$ to come with finite sample guarantees on the false positive rate. In particular, given $\alpha \in \mathbb{R}_{>0}$, if $\mathcal{S} = \bar{\mathcal{T}}_{w,t}$, we want to ensure

$$\mathbb{P}_{S_{w,t} \sim \mathcal{S}^w, T_{w,t} \sim \mathcal{T}_{t-w+1:t}} \left[ \hat{f}(S_{w,t}, T_{w,t}; \alpha) = 0 \right] \geq 1 - \alpha.$$

Ideally, we also want to provide finite sample bounds on the false negative rate; however, for classifier tests, we can only do so under additional assumptions about the model family used to try and distinguish $\mathcal{S}$ and $\bar{\mathcal{T}}_{w,t}$. Intuitively, we assume that (i) the model family has bounded complexity (e.g., Rademacher complexity), and (ii) some model exists in the family that achieves nontrivial accuracy at distinguishing $\mathcal{S}$ and $\bar{\mathcal{T}}_{w,t}$. Then, our goal is to ensure that if $\mathcal{S} \neq \bar{\mathcal{T}}_{w,t}$, we have

$$\mathbb{P}_{S_{w,t} \sim \mathcal{S}^w, T_{w,t} \sim \mathcal{T}_{t-w+1:t}} \left[ \hat{f}(S_{w,t}, T_{w,t}; \alpha) = 1 \right] \geq 1 - M(\alpha, w)$$

for some function $M(\alpha, w)$ that depends on the model family.

### 3.2 ALGORITHM OVERVIEW

Next, we describe our two-sample test. We build on classifier two-sample test (C2ST) (Lopez-Paz & Oquab, 2017; Kim et al., 2021). The idea is to train a classifier $\hat{g}_t$ to try and distinguish $S_{w,t}$

---

**Algorithm 1** Sequential Calibrated Classifier Two-Sample Test

---

1: **Input:** significance level $\alpha$, window size $w$
2: **for** each time step $t$ **do**
3:     Draw examples $x_t \sim \mathcal{S}, x'_t \sim \mathcal{T}_t$
4:     Predict $\hat{y}_t = \hat{g}_t(x_t)$ and $\hat{y}'_t = \hat{g}_t(x'_t)$                    ($\triangleright$) Source-target prediction
5:     Detect covariate shift if $0.5 \notin \Theta_{\text{CP}}(2w\hat{\mu}_{w,t}, 2w; \alpha)$   ($\triangleright$) Calibrated covariate shift detection
6:     Update $\hat{g}_t$ using $(x_t, 0)$ and $(x'_t, 1)$              ($\triangleright$) Online source-target classifier update
7: **end for**

---

from $T_{w,t}$. Intuitively, if $\mathcal{S}$ and $\bar{\mathcal{T}}_{w,t}$ are different distributions, then $\hat{g}_t$ should achieve nontrivial accuracy at distinguishing $S_{w,t}$ from $T_{w,t}$ (assuming the model family is sufficiently expressive). Alternatively, if $\mathcal{S} = \bar{\mathcal{T}}_{w,t}$, then $\hat{g}_t$ necessarily achieves a trivial expected accuracy of $1/2$.

In particular, the accuracy of $\hat{g}_t$ can be used as a test statistic for the two-sample test. To choose the cutoff for rejecting the null hypothesis, we use the Clopper-Pearson (CP) interval (Clopper & Pearson, 1934) to construct an interval that contains the true accuracy $\hat{g}_t$ with high probability based on the accuracy of $\hat{g}_t$ on a test set. More precisely, the CP interval is an exact confidence interval around the empirical estimate of the mean of a Bernoulli random variable. Letting $z_1, ..., z_n \sim$ Bernoulli($\mu^*$) be i.i.d. samples from a Bernoulli distribution with true mean $\mu^*$, the (unnormalized) estimate of its mean $n \cdot \hat{\mu}(z_{1:n}) = \sum_{i=1}^n z_i$ has distribution Binomial($n, \mu^*$). Then, the CP interval $\Theta_{\text{CP}}(s, n; \alpha) \subseteq [0, 1]$ is an interval around $\hat{\mu}$ containing $\mu^*$ with probability at least $1 - \alpha$, *i.e.*,

$$\mathbb{P}_{s \sim \text{Binomial}(n, \mu^*)}[\mu^* \in \Theta_{\text{CP}}(s, n; \alpha)] \geq 1 - \alpha, \tag{1}$$

where the probability is taken over $s$, $\alpha$ is a given confidence level, and $\Theta_{\text{CP}}$ is a function of the Binomial random variable $s = n \cdot \hat{\mu}(z_{1:n})$. The CP interval is concretely defined by

$$\Theta_{\text{CP}}(s, n; \alpha) := \left[\inf\left\{\theta \,\Big|\, F(n - s; n, 1 - \theta) \geq \frac{\alpha}{2}\right\}, \sup\left\{\theta \,\Big|\, F(s; n, \theta) \geq \frac{\alpha}{2}\right\}\right],$$

where $F(s; n, \theta)$ is the cumulative distribution function (CDF) of Binomial($n, \theta$). To compute the CP interval, we can use the following equivalent formula:

$$\Theta_{\text{CP}}(s, n; \alpha) = \left[Q\left(\frac{\alpha}{2}; s, n - s + 1\right), Q\left(1 - \frac{\alpha}{2}; s + 1, n - s\right)\right],$$

where $Q(p, a, b)$ is the $p$th quantile of a Beta distribution with parameters $a, b$ (Hartley & Fitch, 1951; Brown et al., 2001). Our algorithm uses the CP interval to determine whether the accuracy of $\hat{g}_t$ is nontrivial, *i.e.*, $> 1/2$. In particular, the accuracy of $\hat{g}_t$ is the mean of the Bernoulli random variable $\mathbb{1}(\hat{g}_t(x) = y)$, where $y$ is the ground truth indicating whether $x$ is from $\mathcal{S}$ or $\bar{\mathcal{T}}_{w,t}$. Then, our algorithm rejects if the CP interval does not contain $1/2$, since this condition implies that the accuracy of $\hat{g}_t$ does not equal $1/2$ with high probability. We describe this step in detail below.

The key challenge is what data to use as the test dataset to estimate the accuracy of $\hat{g}_t$. The traditional strategy is to split the available data into two parts: one to train $\hat{g}_t$ and a second held-out test set to estimate its accuracy (Lopez-Paz & Oquab, 2017; Kim et al., 2021). However, this approach reduces sample efficiency, which is problematic in our setting since we often want to $w$ to be small.

To address this challenge, our algorithm exploits the conditional independence structure of classifier predictions. In particular, as described below, our algorithm uses each example $x_t$ to evaluate the accuracy of $\hat{g}_t$ *before* using it to train $\hat{g}_t$. In the next section, we prove that this strategy maintains the independence of our estimate of the accuracy of $\hat{g}_t$ (Lemma 1), and that as a consequence, our algorithm satisfies the desired false positive rate (for a single step $t$).

### 3.3 ALGORITHM DETAILS

**Sequential detection algorithm.** At each time step $t$, we observe a source sample $x_t \sim \mathcal{S}$ and a target sample $x'_t \sim \mathcal{T}_t$. By using these current samples and previous samples, we detect covariate shifts by updating the source-target classifier in online learning. In particular, our algorithm consists of three steps: (1) source-target prediction, (2) covariate shift detection, and (3) online source-target classifier update. The following and Algorithm 1 include details.

**Step 1. Source-target prediction.** We predict source-target labels on the current samples $x_t$ and $x'_t$ using the current source-target classifier $\hat{g}_t$. In particular, we denote prediction on the source sample $x_t$ by $\hat{y}_t$, *i.e.*, $\hat{y}_t = \hat{g}_t(x_t)$, and denote prediction on the target sample $x'_t$ by $\hat{y}'_t$, *i.e.*, $\hat{y}'_t = \hat{g}_t(x'_t)$. These predictions and previous predictions are used in covariate shift detection in the following step.

**Step 2. Calibrated covariate shift detection.** Let $\mathcal{Q}_{w,t}$ be a distribution over $\mathcal{X} \times \{0,1\}$, where

$$\mathcal{Q}_{w,t}(x,y) := \frac{1}{2} \cdot \mathcal{S}(x) \cdot \mathbb{1}(y=0) + \frac{1}{2} \cdot \bar{\mathcal{T}}_{w,t}(x) \cdot \mathbb{1}(y=1).$$

Then, $z = \mathbb{1}(\hat{g}_t(x) = y)$ is a Bernoulli random variable with distribution Bernoulli$(\mu^*_{w,t})$, where

$$\mu^*_{w,t} = \mathbb{P}_{(x,y) \sim \mathcal{Q}_{w,t}}[\hat{g}_t(x) = y]$$

is the accuracy of $\hat{g}$ at distinguishing whether an example $x$ is from distribution $\mathcal{S}$ or $\bar{\mathcal{T}}_{w,t}$. The unbiased empirical estimate of this accuracy is denoted by

$$\hat{\mu}_{w,t} = \frac{1}{2w} \sum_{i=t-w+1}^{t} \left( \mathbb{1}\left(\hat{y}_i = y_i\right) + \mathbb{1}\left(\hat{y}'_i = y'_i\right) \right).$$

In fact, $2w\hat{\mu}_{w,t}$ is a Binomial random variable with Binomial$(2w, \mu^*_{w,t})$; thus, the accuracy $\mu^*_{w,t}$ can be estimated by the Clopper-Pearson (CP) interval $\Theta_{\mathrm{CP}}(2w\hat{\mu}_{w,t}, 2w; \alpha)$ that includes the unknown parameter $\mu^*_{w,t}$ with high probability, *i.e.*,

$$\mathbb{P}\left[\mu^* \in \Theta_{\mathrm{CP}}(2w\hat{\mu}_{w,t}, 2w; \alpha)\right] \geq 1 - \alpha.$$

This property can be used for checking the accuracy of $\hat{g}_t$ might be $1/2$. In particular, our algorithm returns "covariate shift" if $1/2 \notin \Theta_{\mathrm{CP}}(2w\hat{\mu}_{w,t}, 2w; \alpha)$, and "no covariate shift" otherwise, *i.e.*

$$\hat{f}(S_{w,t}, T_{w,t}; \alpha) = \mathbb{1}\left( \frac{1}{2} \notin \Theta_{\mathrm{CP}}\left(2w \cdot \hat{\mu}_{w,t}, 2w; \alpha\right) \right).$$

Here, the Clopper-Pearson interval calibrates the empirical accuracy $\hat{\mu}_{w,t}$ using the property of the Binomial distribution.

**Step 3. Online source-target classifier update.** Finally, we update a binary classifier $\hat{g}_t$ using new training examples based on the source and target samples, *i.e.*, $(x_t, 0)$ and $(x'_t, 1)$. In general, $\hat{g}_t$ can be any model; we consider it to be a neural network, in which case we can update its parameters using stochastic gradient descent with respect to the cross entropy loss.

## 4 THEORETICAL GUARANTEES

In this section, we describe our finite sample bounds on the false positive and false negative rates of our covariate shift detector $\hat{f}$; the key to have valid bounds is proving the independence on predictions $\hat{y}_1, \ldots, \hat{y}_t$ (and $\hat{y}'_1, \ldots, \hat{y}'_t$) to have a valid Clopper-Pearson interval, since they are seemingly dependent through the online learned classifier $\hat{g}_t$. First, our key result shows that our estimate of the accuracy of $\hat{g}_t$ is valid—*i.e.*, the predictions $\hat{y}_i, \ldots, \hat{y}_j$ are conditionally independent (see Appendix A.1 for a proof), thus the accuracy is the parameter of the Binomial distribution:

**Lemma 1.** *If $x_i, \ldots, x_j$ are independent for any $i, j \in \mathbb{N}$ where $i < j$, $\hat{y}_i, \ldots, \hat{y}_j$ are conditionally independent given $\hat{g}_i, \ldots, \hat{g}_{j-1}$.*

Our next result says that our algorithm ensures the desired bound $\alpha$ on the false positive rate (i.e., $\hat{f}$ says "covariate shift" when there is no covariate shift). To this end, we exploit the following observation that *any* source-target classifier makes the expected accuracy of $1/2$ if there is no covariate shift. Intuitively, if $\mathcal{S} = \bar{\mathcal{T}}_{w,t}$, source-target classification is impossible (Lopez-Paz & Oquab, 2017; Liu et al., 2020); we include this lemma for completeness (see Appendix A.2 for a proof):

**Lemma 2.** *If $\mathcal{S} = \bar{\mathcal{T}}_{w,t}$, we have $\mu^*_{w,t} = 1/2$ for any source-target classifier $\hat{g}_t$.*

Since the expected accuracy of $\hat{g}_t$ is $1/2$ *regardless of* how we design and learn $\hat{g}_t$, and how many samples are used to learn $\hat{g}_t$, the Clopper-Pearson interval includes the true accuracy with high probability; thus the false positive rate of the proposed covariate shift detector $\hat{f}$ is effectively controlled by the confidence level of the Clopper-Pearson interval, as follows (see Appendix A.3 for a proof):

**Theorem 3** (Bound on false positive rate). *If $\mathcal{S} = \bar{\mathcal{T}}_{w,t}$, then for any $\hat{g}_t$, we have*

$$\mathbb{P}_{(S_{w,t},T_{w,t})\sim\mathcal{S}^w\times\mathcal{T}_{t-w+1:t}}\left[\hat{f}(S_{w,t},T_{w,t};\alpha)=0\right]\geq 1-\alpha. \quad (2)$$

Our next result provides a bound on the false negative rate; we first observe that the Clopper-Pearson interval is included in the interval by the Hoeffding's bound. Intuitively, the Clopper-Pearson interval represents a lower and upper bound of the expected accuracy given an empirical accuracy tailored to a Bernoulli random variable; the Hoeffding's bound can similarly bound the mean but in a more general setup. Thus, the Clopper-Pearson interval can be smaller (see Appendix A.4 for a proof).

**Lemma 4.** *Let $s \sim Binomial(n, p)$ and $F(s; n, p)$ is the CDF of $Binomial(n, p)$; we have*

$$\frac{s}{n}-\sqrt{\frac{\ln\frac{2}{\alpha}}{2n}}\leq\inf\left\{\theta\ \Big|\ F(n-s;n,1-\theta)\geq\frac{\alpha}{2}\right\}\ and\ \sup\left\{\theta\ \Big|\ F(s;n,\theta)\geq\frac{\alpha}{2}\right\}\leq\frac{s}{n}+\sqrt{\frac{\ln\frac{2}{\alpha}}{2n}}.$$

Leveraging this, we have the following bound on false negative rate (see Appendix A.5 for a proof).

**Theorem 5** (Bound on false negative rate). *Assume $\hat{g}_t$ achieves nontrivial accuracy, i.e., $\mu^*_{w,t} \geq 1/2 + \epsilon$, where $\epsilon \in (0, 1/2]$, is the accuracy at distinguishing $\mathcal{S}$ and $\bar{\mathcal{T}}_{w,t}$. Let $a(w,\alpha) := 2w(1/2 + \sqrt{\log(2/\alpha)/4w})$ and $b(w,\alpha) := 2w(1/2 - \sqrt{\log(2/\alpha)/4w})$. If $\mathcal{S} \neq \bar{\mathcal{T}}_{w,t}$ and $w - 1 - \lfloor\sqrt{w\log(2/\alpha)}\rfloor \geq 0$, then we have*

$$\mathbb{P}\left[\hat{f}(S_{w,t},T_{w,t};\alpha)=1\right]\geq F\left(2w-\lfloor a(w,\alpha)+1\rfloor;2w,\frac{1}{2}-\epsilon\right)+F\left(\lceil b(w,\alpha)-1\rceil;2w,1\right). \quad (3)$$

In the false negative bound, the first term is dominant and increases as $w$ increases, which implies the sample size needs to be increased to have a powerful shift detector; the condition on $w$ suggests that the bound is valid when $w \geq 201$ given $\alpha = 0.01$. We note that the assumption $L(\hat{g}_t) := 1 - \mu^* \leq 1/2 - \epsilon$ can be achieved under standard conditions. For instance, assume that the model family $\mathcal{G}$ of source-target classifiers has finite VC dimension (*i.e.*, $\text{VC}(\mathcal{G}) < \infty$), and that the optimal model $g^* \in \mathcal{G}$ has nontrivial inaccuracy $L(g^*) = 1/2 - \xi$ for some $\xi \in \mathbb{R}_{>0}$; then, with probability at least $1 - \delta$ with respect to $S_{w,t}$ and $T_{w,t}$ and letting $m = 2w$, we have

$$L(\hat{g}_t) \leq L(g^*) + 4\sqrt{\frac{\text{VC}(\mathcal{G})(\log(2m)+1)}{m}} + \sqrt{\frac{\log(2/\delta)}{m}}$$

$$\leq \frac{1}{2} - \underbrace{\left(\xi - 4\sqrt{\frac{\text{VC}(\mathcal{G})(\log(2m)+1)}{m}} - \sqrt{\frac{\log(2/\delta)}{m}}\right)}_{=:\epsilon},$$

where the second term (which we have taken to be $\epsilon$) satisfies $\epsilon > 0$ for sufficiently large $m$.

## 5 EXPERIMENTS

We evaluate the effectiveness of our algorithm at detecting both natural and synthetic covariate shifts of varying forms (e.g., gradual shifts and multiple shifts back and forth), showing that it significantly outperforms natural baselines.

### 5.1 EXPERIMENT SETUP

**Baselines.** We compare our algorithm to three baselines: two of them differ in the way they use the samples at each time step, the third uses Wald's sequential likelihood test (Wald, 1945). For the first two baselines, while our approach uses all samples to construct the CP interval around the accuracy of the source-target classifier $\hat{g}_t$ as well as to train $\hat{g}_t$, the baseline instead constructs a held-out test set using every $H^{th}$ sample. Then, only this held-out test set is used to compute the CP interval, and only the remaining samples are used to train $\hat{g}_t$. In our experiments, we used values of $H \in \{2, 5\}$, denoted H2, H5, respectively. For Wald's test, we consider the Bernoulli random variable with a probability $p$ indicating whether the prediction of the source-target classifier is correct for the given

Table 1: Scenario description for experiments. (a) "M-shift" is Multiple shift, (b) "GI-shift" is gradually increasing shift, and (c) "GID-shift" is gradually increasing-then-decreasing shift.

(a) M-shift

| Start position | Description | Prob. |
|---|---|---|
| 0% | No shift | 0.0 |
| 25% | Shift | 1.0 |
| 50% | No shift | 0.0 |
| 75% | Shift | 1.0 |

(b) GI-shift

| Start position | Description | Prob. |
|---|---|---|
| 0% | No shift | 0.0 |
| 20% | Shift | 0.2 |
| 40% | Shift | 0.4 |
| 60% | Shift | 0.6 |
| 80% | Shift | 0.8 |

(c) GID-shift

| Start position | Description | Prob. |
|---|---|---|
| 0 | No shift | 0.0 |
| 20% | Shift | 0.4 |
| 40% | Shift | 0.8 |
| 60% | Shift | 0.4 |
| 80% | No shift | 0.0 |

sample. The hypothesis test is $H_0 : p = 0.5$ vs. $H_1 : p = 0.5 + \epsilon$, where $\epsilon = 0.2$ in our experiments; we restart the test each time it makes a decision.

This baseline is essentially the online version of an existing classifier two-sample test (C2ST) (Kim et al., 2021; Lopez-Paz & Oquab, 2017), which splits the (fixed) training dataset into a training set to train $\hat{g}_t$ and a held-out test set to estimate the accuracy of $\hat{g}_t$; thus, $H$ controls the tradeoff between the number of examples in the training set and held-out test set.

**Source-target classifier.** We use a fully-connected neural network with a single hidden layer (with 128 hidden units) and with the ReLU activation functions as the source-target classifier $\hat{g}_t$. We use a binary cross-entropy loss for training in conjunction with an SGD optimizer with a learning rate of 0.01 (for natural shift experiments) and 0.001 (for synthetic shift experiments). Finally, since the inputs are ImageNet images (Russakovsky et al., 2015), we use a 2048-dimensional feature vector generated by first running a pretrained ResNet152 model (He et al., 2016) on the images, and then using these features vectors for the covariates of $S_{w,t}$ and $T_{w,t}$.

**Scenarios.** We run each algorithm to test whether the target distribution in the given window is shifted with three different scenarios: multiple shift ("M-shift"), gradually increasing shift ("GI-shift"), and gradually increasing-then-decreasing shift ("GID-shift"). Table 1 describes each scenario. For example, the multiple shift scenario proceeds as follows: (i) it starts with no covariate shift at the beginning; (ii) after observing 25% target samples (*i.e.*, $250^{th}$ samples for natural shift experiments and $2500^{th}$ samples for synthetic shift experiments), covariate shift is applied to all target samples (with probability 1) by adding random perturbations for synthetic shift and by drawing samples from a target distribution for natural shift; (iii) after 50% of target samples, it reverts to no covariate shift; and (iv) finally after observing 75% target samples, the covariate shift is applied to the all target samples. Gradually increasing shift and gradually increasing-then-decreasing shift scenarios start with no covariate shift for the first 20% of target samples; then, covariate shift is applied with some probability $0 < p < 1$ by gradually changing $p$ over time.

**Stream data generation.** For each shift (*i.e.*, natural shift and synthetic shift), we have a source dataset $\mathcal{S}$ and target datasets $\mathcal{T}_t$, from which we randomly draw source and target samples for each time step $t$. In particular, we consider a batch of samples for computational efficiency, where we denote the batch size by $B$; we use $B = 10$ for our experiments. That is, we wait for $B$ samples to be collected from the target distribution before checking for covariate shift and the updating the source-target classifier; then, we begin collecting the next batch. Finally, we evaluate each approach using multiple random repetitions, which we denote by $R$ (the value of $R$ depends on each experiment).

## 5.2 NATURAL SHIFT

**Dataset.** First, we consider a natural shift on ImageNet. To construct such a shift, we consider the subset of dog classes; in particular, 120 of the 1000 of the ImageNet classes are of dogs (Khosla et al., 2011). Then, we randomly select half (*i.e.*, 60) of these classes to be the source dataset, and the other half to be the target dataset; thus, the number of source and target images is 2997 each (after removing duplicated images). As a consequence, the source and target datasets correspond to different dog breeds, which is a kind of natural distribution shift.

**Results.** Figure 1 and Table 2 show results for the natural shift experiment with $w = 10$ and $\alpha = 1\%$. Figure 1 illustrates detection rates of the four algorithms with $R = 100$ repetitions (i.e., the fraction of repetitions that reported "covariate shift" at each step). Table 2a shows the number

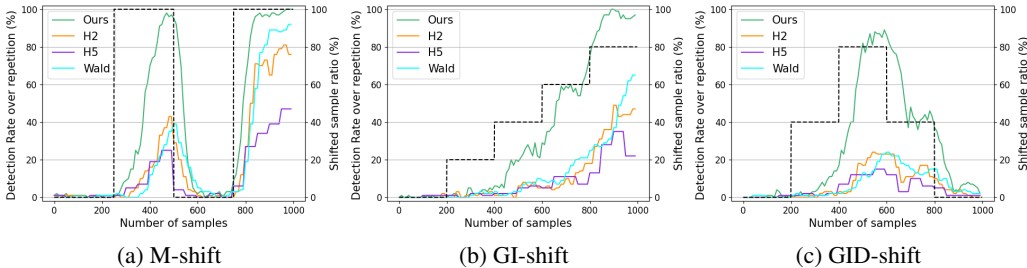

| (a) M-shift | (b) GI-shift | (c) GID-shift |

Figure 1: Detection rate for natural shift with $R = 100$, $w = 10$, $\alpha = 1\%$. The black dashed line indicates shifted sample ratio, *i.e.*, the degree (or probability) of covariate shift.

Table 2: Natural shift results with (a) $w = 10$, $\alpha = 1\%$, and $R = 100$, and (b) $R = 20000$ . In (a), we bold the best algorithm. In (b), we bold values that exceed the desired $\alpha = 1\%$.

(a) Number of samples for detection ($\geq 80\%$)

| Scenario | Algorithm | Natural shift |
|---|---|---|
| M-shift | Ours | **190** |
| | H2 | 720 |
| | H5 | - |
| | Wald | 640 |
| GI-shift | Ours | **620** |
| | H2 | - |
| | H5 | - |
| | Wald | - |
| GID-shift | Ours | **310** |
| | H2 | - |
| | H5 | - |
| | Wald | - |

(b) FPR (%) at selected time

| Scenario | Algorithm | 50 | 100 | 150 | 200 |
|---|---|---|---|---|---|
| M-shift | Ours | 0.27 | 0.53 | 0.73 | 0.77 |
| | H2 | 0.29 | 0.28 | 0.26 | 0.33 |
| | H5 | 0.34 | 0.52 | 0.51 | 0.56 |
| | Wald | 0.60 | 0.47 | 0.27 | 0.27 |
| GI-shift | Ours | 0.21 | 0.60 | 0.76 | 0.83 |
| | H2 | 0.21 | 0.25 | 0.29 | 0.36 |
| | H5 | 0.32 | 0.43 | 0.50 | 0.85 |
| | Wald | 0.78 | 0.57 | 0.29 | 0.22 |
| GID-shift | Ours | 0.30 | 0.53 | 0.70 | 0.95 |
| | H2 | 0.18 | 0.21 | 0.28 | 0.41 |
| | H5 | 0.36 | 0.56 | 0.53 | 0.81 |
| | Wald | 0.77 | 0.58 | 0.34 | 0.23 |

of shifted samples required to reach at least 80% of covariate shift detection rate under the shift. Table 2b shows false positive rate (FPR) after 50, 100, 150, and 200 samples with $R = 20000$ repetitions.

**Discussion.** Figure 1 shows the detection rate of each algorithm as each scenario progresses. In multiple shift (Figure 1a) and gradually increasing-then-decreasing shift (Figure 1c), covariate shift disappears after a certain point, and all algorithms correctly detect this change. However, as shown in Table 2a, our approach always requires fewer samples to detect the shift. While H5 does not achieve 80% detection and H2 and Wald reach 80% only for multiple shift change scenario, our approach always detects covariate shift at a rate higher than 80%. Furthermore, for multiple shift, our algorithm requires fewer than half the number of samples compared to H2. In summary, our algorithm is significantly more sample efficient at detecting covariate shift compared to the baselines, most likely since it utilizes all samples for both training the source-target classifier and constructing the CP interval. For FPR, all algorithms always satisfy the FPR bound (*i.e.*, FPR $\leq \alpha$).

### 5.3 SYNTHETIC SHIFT

**Dataset.** Next, we consider a synthetic shift on ImageNet. In particular, we split the original ImageNet validation set into equal sized source and target datasets. To construct the target dataset, we add synthetic perturbations on original images. We (separately) consider five perturbation types from (Hendrycks & Dietterich, 2019)—in particular, Contrast, Defocus Blur, Elastic Transform, Gaussian Blur, and Gaussian Noise, with five different severity levels.

**Results.** The experiment results are shown in Figure 2 and Table 3 for the experiments with the perturbation severity of 2, window size $w = 10$, and significance level $\alpha = 1\%$. Table 3a shows the number of target samples required by each algorithm to detect the first covariate shift in the detection rate of at least 80%. Table 3b shows the false positive rate (FPR) after 500, 1000, 1500 and 2000 samples for each of the three scenarios. Figure 2 shows the detection rates over multiple repetitions

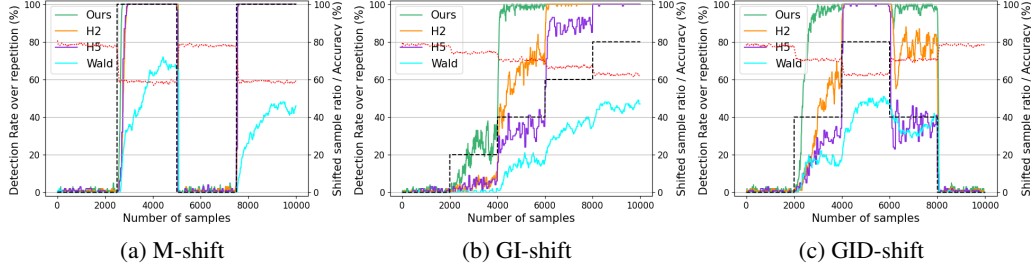

(a) M-shift  (b) GI-shift  (c) GID-shift

Figure 2: Detection rate for synthetic shifts with Gaussian noise perturbation, Severity $= 2$, $R = 100$, $w = 10$, $\alpha = 1\%$. The black dashed line indicates shifted sample ratio, *i.e.*, the degree (or probability) of covariate shift. The red dotted line shows the accuracy of ResNet152 on the source and target samples in the given window.

Table 3: Synthetic shift results with (a) Severity $= 2$, $w = 10$, $\alpha = 1\%$, and $R = 100$, and (b) $R = 20000$. In (a), we bold the best algorithm. In (b), we bold values that exceed the desired $\alpha = 1\%$.

(a) Number of samples for detection

| Scenario | Alg. | Contrast | Defocus Blur | Elastic Transform | Gaussian Blur | Gaussian Noise |
|---|---|---|---|---|---|---|
| M-shift | Ours | **230** | **200** | **220** | **210** | **180** |
| | H2 | 470 | 450 | 450 | 490 | 350 |
| | H5 | 410 | 410 | 410 | 460 | 310 |
| | Wald | - | - | - | - | - |
| GI-shift | Ours | **2100** | **2060** | **2090** | **2070** | **2080** |
| | H2 | 4050 | 3690 | 4050 | 4010 | 3670 |
| | H5 | 4360 | 4110 | 6010 | 4110 | 4110 |
| | Wald | - | - | - | - | - |
| GID-shift | Ours | **880** | **560** | **900** | **720** | **610** |
| | H2 | 2030 | 2010 | 2050 | 2010 | 2010 |
| | H5 | 2060 | 2060 | 2060 | 2060 | 2060 |
| | Wald | - | - | - | - | - |

(b) FPR (%) at selected time

| Scenario | Alg. | 500 | 1000 | 1500 | 2000 |
|---|---|---|---|---|---|
| M-shift | Ours | 0.73 | 1.00 | 0.90 | 1.00 |
| | H2 | 0.40 | 0.46 | 0.46 | 0.50 |
| | H5 | 0.80 | 0.73 | 0.84 | 0.69 |
| | Wald | 0.07 | 0.11 | 0.04 | 0.01 |
| GI-shift | Ours | 0.86 | 0.89 | 0.92 | 0.85 |
| | H2 | 0.62 | 0.54 | 0.60 | 0.50 |
| | H5 | 0.74 | 0.71 | 0.77 | 0.73 |
| | Wald | 0.10 | 0.13 | 0.07 | 0.03 |
| GID-shift | Ours | 0.87 | 0.95 | 0.92 | 0.89 |
| | H2 | 0.51 | 0.57 | 0.51 | 0.53 |
| | H5 | 0.73 | 0.75 | 0.85 | 0.89 |
| | Wald | 0.09 | 0.08 | 0.04 | 0.02 |

for each of the three scenarios using the Gaussian noise perturbation. Results for other perturbation types and severities are shown in Appendix D.

**Discussion.** As can be seen, our approach outperforms the baselines in terms of sample efficiency for the covariate shift detection as was the case of the natural shift. Our algorithm requires about half as many samples before detecting covariate shift compared to the baselines. In terms of FPR, our approach always satisfies the FPR bound. Finally, Figure 2 shows the accuracy drop with the shifted samples. In particular, the red dotted line shows the accuracy of ResNet152 on the examples in the source and target samples of the given window; as can be seen, the accuracy decreases as the degree of the shift increases. Covariate shift detection can be successfully used to notify a user that an accuracy drop may have occurred.

# 6 CONCLUSION

We have proposed a novel covariate shift detection algorithm, which uses a classifier two-sample test to check whether the current test examples differ in distribution compared to the training examples. Our approach ensures sample efficiency by avoiding the need to split the dataset into a training set and a held-out test set, and instead using all the data to both train the source-target discriminator and to evaluate its accuracy. We prove that even with this optimization, our approach provides finite sample guarantees on the false positive rate at a desired level; we also prove bounds on the false negative rate under a mild conditions on the trained classifier. Finally, we empirically demonstrate that our proposed algorithm is significantly more sample efficient compared to a natural baseline that uses a held-out test set in terms of detecting both natural and synthetic shifts on ImageNet.

**Reproducibility Statement.** For our empirical results, we stated our algorithm in Algorithm 1, hyperparameters in Section 5.1, and dataset setups in Section 5.2 and 5.3. We have included the source code in the supplement for reproducing the experimental results. For our theory, we have included all proofs in Appendix A.

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

# A PROOFS

## A.1 PROOF OF LEMMA 1

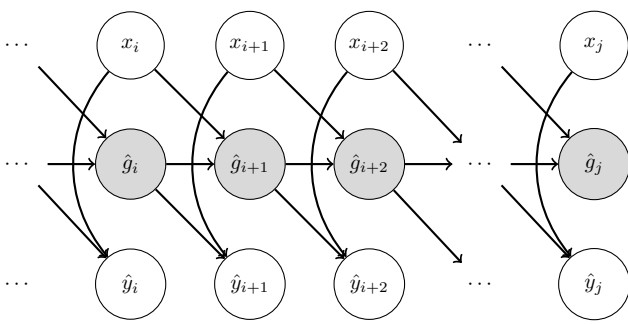

Figure 3: The dependency structure of random variables.

Figure 3 represents the graphical model over random variables, where observed random variables are colored in gray. We prove the conditional independence using the d-separation (also called the Bayes ball algorithm) (Bishop, 2006), which is a set of rules that can determine the conditional dependency between two random variables based on the graphical model and observed random variables. In particular, $\hat{y}_{i+2}$ is conditionally independent to $\hat{y}_k$ for all $k \leq i+1$ since the path to $\hat{y}_k$ is blocked by $\hat{g}_{i+1}$ (*i.e.*, $\hat{g}_{i+1}$ is observed). Similarly, $\hat{y}_{i+2}$ is conditionally independent to $\hat{y}_k$ for all $k \geq i+3$. This proves the claim. □

## A.2 PROOF OF LEMMA 2

For any source-target classifier $\hat{g}_t$, if $\mathcal{S} = \hat{\mathcal{T}}_{w,t}$, the following holds:

$$
\begin{aligned}
\mu^*_{w,t} &= \mathbb{P}_{(x,y)\sim\mathcal{Q}_{w,t}}\left[\hat{g}_t(x) = y\right] \\
&= \int \sum_{y\in\{0,1\}} \mathbb{1}\left(\hat{g}_t(x) = y\right)\mathcal{Q}_{w,t}(x,y)\mathrm{d}x \\
&= \int \sum_{y\in\{0,1\}} \mathbb{1}\left(\hat{g}_t(x) = y\right)\left(\frac{1}{2}\cdot\mathcal{S}(x)\cdot\mathbb{1}(y=0) + \frac{1}{2}\cdot\bar{\mathcal{T}}_{w,t}(x)\cdot\mathbb{1}(y=1)\right)\mathrm{d}x \\
&= \frac{1}{2}\int \sum_{y\in\{0,1\}} \mathbb{1}\left(\hat{g}_t(x) = y\right)\mathcal{S}(x)\mathbb{1}(y=0) + \sum_{y\in\{0,1\}} \mathbb{1}\left(\hat{g}_t(x) = y\right)\bar{\mathcal{T}}_{w,t}(x)\mathbb{1}(y=1)\mathrm{d}x \\
&= \frac{1}{2}\int \mathbb{1}\left(\hat{g}_t(x) = 0\right)\mathcal{S}(x) + \mathbb{1}\left(\hat{g}_t(x) = 1\right)\bar{\mathcal{T}}_{w,t}(x)\mathrm{d}x \\
&= \frac{1}{2}\int \mathbb{1}\left(\hat{g}_t(x) = 0\right)\mathcal{S}(x) + \mathbb{1}\left(\hat{g}_t(x) = 1\right)\mathcal{S}(x)\mathrm{d}x \\
&= \frac{1}{2}\int \left(\mathbb{1}\left(\hat{g}_t(x) = 0\right) + \mathbb{1}\left(\hat{g}_t(x) = 1\right)\right)\mathcal{S}(x)\mathrm{d}x \\
&= \frac{1}{2}\int \mathcal{S}(x)\mathrm{d}x \\
&= \frac{1}{2},
\end{aligned}
$$

where the sixth equality holds since $\mathcal{S} = \hat{\mathcal{T}}_{w,t}$; the claim follows. □

### A.3    PROOF OF THEOREM 3

Denote the event that $\mathbb{P}_{(x,y)\sim\mathcal{Q}_{w,t}}\left[\hat{g}_t(x)=y\right]=1/2$ by $E$, and let $s_{w,t}=2w\hat{\mu}_{w,t}$. Then, we have

$$
\mathbb{P}_{S_{w,t},T_{w,t}}\left[\hat{f}(S_{w,t},T_{w,t};\alpha)=0\right]
$$

$$
=\mathbb{P}_{S_{w,t},T_{w,t}}\left[\left(\frac{1}{2}\in\Theta_{\mathrm{CP}}(s_{w,t},2w;\alpha)\right)\wedge\left(\mathbb{P}_{x,y}\left[\hat{g}_t(x)=y\right]=\frac{1}{2}\right)\right]
$$

$$
=\mathbb{P}_{S_{w,t},T_{w,t}}\left[E\right]\mathbb{P}_{S_{w,t},T_{w,t}}\left[\frac{1}{2}\in\Theta_{\mathrm{CP}}(s_{w,t},2w;\alpha)\ \middle|\ E\right]
$$

$$
=\mathbb{P}_{S_{w,t},T_{w,t}}\left[\frac{1}{2}\in\Theta_{\mathrm{CP}}(s_{w,t},2w;\alpha)\ \middle|\ E\right]
$$

$$
\geq 1-\alpha,
$$

where the first equality holds since $\mathcal{S}=\hat{\mathcal{T}}_{w,t}$ and by Lemma 2, the third equality holds by Lemma 2, and the last inequality holds by the property of the Clopper-Pearson interval and Lemma 1.    □

### A.4    PROOF OF LEMMA 4

We use the tail bound of the binomial distribution using the Hoeffding's inequality—*i.e.*

$$
F(s;n,p)\leq\exp\left\{-2n\left(p-\frac{s}{n}\right)^2\right\}.
$$

For the upper bound of the upper Clopper-Pearson interval, we have

$$
\sup\left\{\theta\ \middle|\ F(s;n,\theta)\geq\frac{\alpha}{2}\right\}\leq\sup\left\{\theta\ \middle|\ \exp\left\{-2n\left(\theta-\frac{s}{n}\right)^2\right\}\geq\frac{\alpha}{2}\right\}
$$

$$
=\sup\left\{\theta\ \middle|\ \frac{s}{n}-\sqrt{\frac{\ln\frac{2}{\alpha}}{2n}}\leq\theta\leq\frac{s}{n}+\sqrt{\frac{\ln\frac{2}{\alpha}}{2n}}\right\}
$$

$$
=\frac{s}{n}+\sqrt{\frac{\ln\frac{2}{\alpha}}{2n}}. \tag{4}
$$

For the lower bound of the lower Clopper-Pearson interval, we have

$$
\inf\left\{\theta\ \middle|\ F(n-s;n,1-\theta)\geq\frac{\alpha}{2}\right\}\geq\inf\left\{\theta\ \middle|\ \exp\left\{-2n\left(\theta-\frac{s}{n}\right)^2\right\}\geq\frac{\alpha}{2}\right\}
$$

$$
=\inf\left\{\theta\ \middle|\ \frac{s}{n}-\sqrt{\frac{\ln\frac{2}{\alpha}}{2n}}\leq\theta\leq\frac{s}{n}+\sqrt{\frac{\ln\frac{2}{\alpha}}{2n}}\right\}
$$

$$
=\frac{s}{n}-\sqrt{\frac{\ln\frac{2}{\alpha}}{2n}}. \tag{5}
$$

Finally, (4) and (5) imply the claim.    □

### A.5    PROOF OF THEOREM 5

Let the lower and upper bound of the Clopper-Pearson interval $\Theta_{\mathrm{CP}}$ be $\underline{\Theta}_{\mathrm{CP}}$ and $\overline{\Theta}_{\mathrm{CP}}$, respectively. Recall that we denote the CDF of a binomial distribution $\mathrm{binomial}(n,p)$ by $F(s;n,p)$. Then, we

have

$$\mathbb{P}_{S_{w,t},T_{w,t}} \left[ \hat{f}(S_{w,t}, T_{w,t}; \alpha) = 1 \right]$$

$$= \mathbb{P}_{S_{w,t},T_{w,t}} \left[ \frac{1}{2} \notin \Theta_{\mathrm{CP}}(2w\mu^*_{w,t}, 2w; \alpha) \right]$$

$$= \mathbb{P}_{S_{w,t},T_{w,t}} \left[ \left( \mu^*_{w,t} < \frac{1}{2} + \epsilon \vee \mu^*_{w,t} \geq \frac{1}{2} + \epsilon \right) \wedge \left( \frac{1}{2} \notin \Theta_{\mathrm{CP}}(2w\mu^*_{w,t}, 2w; \alpha) \right) \right]$$

$$= \mathbb{P}_{S_{w,t},T_{w,t}} \left[ \left( \mu^*_{w,t} \geq \frac{1}{2} + \epsilon \right) \wedge \left( \frac{1}{2} \notin \Theta_{\mathrm{CP}}(2w\mu^*_{w,t}, 2w; \alpha) \right) \right] \tag{6}$$

$$= \mathbb{P}_{S_{w,t},T_{w,t}} \left[ \mu^*_{w,t} \geq \frac{1}{2} + \epsilon \right] \mathbb{P}_{S_{w,t},T_{w,t}} \left[ \frac{1}{2} \notin \Theta_{\mathrm{CP}}(2w\mu^*_{w,t}, 2w; \alpha) \; \middle| \; \mu^*_{w,t} \geq \frac{1}{2} + \epsilon \right]$$

$$= \mathbb{P}_{S_{w,t},T_{w,t}} \left[ \frac{1}{2} \notin \Theta_{\mathrm{CP}}(2w\mu^*_{w,t}, 2w; \alpha) \; \middle| \; \mu^*_{w,t} \geq \frac{1}{2} + \epsilon \right] \tag{7}$$

$$= \mathbb{P}_{S_{w,t},T_{w,t}} \left[ \underline{\Theta}_{\mathrm{CP}}(2w\mu^*_{w,t}, 2w; \alpha) > \frac{1}{2} \vee \overline{\Theta}_{\mathrm{CP}}(2w\mu^*_{w,t}, 2w; \alpha) < \frac{1}{2} \; \middle| \; \mu^*_{w,t} \geq \frac{1}{2} + \epsilon \right]$$

$$= \mathbb{P}_{S_{w,t},T_{w,t}} \left[ \underline{\Theta}_{\mathrm{CP}}(2w\mu^*_{w,t}, 2w; \alpha) > \frac{1}{2} \; \middle| \; \mu^*_{w,t} \geq \frac{1}{2} + \epsilon \right]$$

$$+ \mathbb{P}_{S_{w,t},T_{w,t}} \left[ \overline{\Theta}_{\mathrm{CP}}(2w\mu^*_{w,t}, 2w; \alpha) < \frac{1}{2} \; \middle| \; \mu^*_{w,t} \geq \frac{1}{2} + \epsilon \right],$$

where (6) and (7) hold due to $\mathbb{P}_{S_{w,t},T_{w,t}}[\mu^*_{w,t} < 1/2 + \epsilon] = 0$ and $\mathbb{P}_{S_{w,t},T_{w,t}}[\mu^*_{w,t} \geq 1/2 + \epsilon] = 1$ from the assumption on $\hat{g}_t$ and $\mathcal{S} \neq \mathcal{T}$, respectively.

By Lemma 4, the first term is lower bounded as follows:

$$\mathbb{P}_{S_{w,t},T_{w,t}} \left[ \underline{\Theta}_{\mathrm{CP}}(2w\mu^*_{w,t}, 2w; \alpha) > \frac{1}{2} \; \middle| \; \mu^*_{w,t} \geq \frac{1}{2} + \epsilon \right]$$

$$\geq \mathbb{P}_{S_{w,t},T_{w,t}} \left[ \hat{\mu}_{w,t} - \sqrt{\frac{\ln \frac{2}{\alpha}}{4w}} > \frac{1}{2} \; \middle| \; \mu^*_{w,t} \geq \frac{1}{2} + \epsilon \right]$$

$$= \mathbb{P}_{S_{w,t},T_{w,t}} \left[ 2w\hat{\mu}_{w,t} > a(w, \alpha) \; \middle| \; \mu^*_{w,t} \geq \frac{1}{2} + \epsilon \right]$$

$$= \mathbb{P}_{S_{w,t},T_{w,t}} \left[ 2w\hat{\mu}_{w,t} \geq \lfloor a(w, \alpha) + 1 \rfloor \; \middle| \; \mu^*_{w,t} \geq \frac{1}{2} + \epsilon \right]$$

$$\geq F \left( 2w - \lfloor a(w, \alpha) + 1 \rfloor; 2w, \frac{1}{2} - \epsilon \right), \tag{8}$$

where the last inequality holds since the binomial parameter $\frac{1}{2} - \epsilon$ makes the CDF $F$ smallest.

Similarly, the second term is lower bounded as follows:

$$\mathbb{P}_{S_{w,t},T_{w,t}} \left[ \overline{\Theta}_{\mathrm{CP}}(2w\mu^*_{w,t}, 2w; \alpha) < \frac{1}{2} \; \middle| \; \mu^*_{w,t} \geq \frac{1}{2} + \epsilon \right]$$

$$\geq \mathbb{P}_{S_{w,t},T_{w,t}} \left[ \hat{\mu}_{w,t} + \sqrt{\frac{\ln \frac{2}{\alpha}}{4w}} < \frac{1}{2} \; \middle| \; \mu^*_{w,t} \geq \frac{1}{2} + \epsilon \right]$$

$$= \mathbb{P}_{S_{w,t},T_{w,t}} \left[ 2w\hat{\mu}_{w,t} < b(w, \alpha) \; \middle| \; \mu^*_{w,t} \geq \frac{1}{2} + \epsilon \right]$$

$$= \mathbb{P}_{S_{w,t},T_{w,t}} \left[ 2w\hat{\mu}_{w,t} \leq \lceil b(w, \alpha) - 1 \rceil \; \middle| \; \mu^*_{w,t} \geq \frac{1}{2} + \epsilon \right]$$

$$\geq F \left( \lceil b(w, \alpha) - 1 \rceil; 2w, 1 \right), \tag{9}$$

where the last inequality holds since the binomial parameter $\mu^*_{w,t} = 1$ makes the CDF $F$ smallest.

The claim follows by combining (8) and (9). □

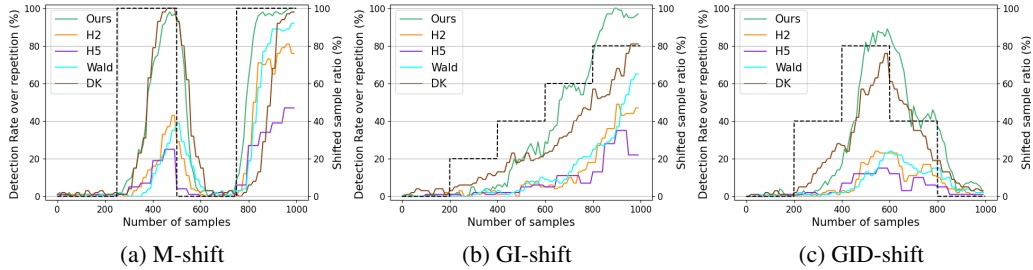

|          | (a) M-shift | (b) GI-shift | (c) GID-shift |

Figure 4: Detection rate for natural shift with $R = 100$, $w = 10$, $\alpha = 1\%$. Deep Kernel result is included. The black dashed line indicates shifted sample ratio, *i.e.*, the degree (or probability) of covariate shift.

Table 4: Natural shift results with (a) $w = 10$, $\alpha = 1\%$, and $R = 100$, and (b) $R = 20000$. In (a), we bold the best algorithm. In (b), we bold values that exceed the desired $\alpha = 1\%$.

<table>
<tr><td colspan="3">(a) Number of samples for detection ($\geq 80\%$)</td><td colspan="5">(b) FPR (%) at selected time</td></tr>
</table>

| Scenario | Algorithm | Natural shift | | Scenario | Algorithm | 50 | 100 | 150 | 200 |
|----------|-----------|---------------|---|----------|-----------|-----|-----|-----|-----|
| M-shift | Ours | 190 | | M-shift | Ours | 0.27 | 0.53 | 0.73 | 0.77 |
|  | H2 | 720 | |  | H2 | 0.29 | 0.28 | 0.26 | 0.33 |
|  | H5 | - | |  | H5 | 0.34 | 0.52 | 0.51 | 0.56 |
|  | Wald | 640 | |  | Wald | 0.60 | 0.47 | 0.27 | 0.27 |
|  | DK | **180** | |  | DK | **1.69** | **2.31** | **2.16** | **2.54** |
| GI-shift | Ours | **620** | | GI-shift | Ours | 0.21 | 0.60 | 0.76 | 0.83 |
|  | H2 | - | |  | H2 | 0.21 | 0.25 | 0.29 | 0.36 |
|  | H5 | - | |  | H5 | 0.32 | 0.43 | 0.50 | 0.85 |
|  | Wald | - | |  | Wald | 0.78 | 0.57 | 0.29 | 0.22 |
|  | DK | 770 | |  | DK | **2.11** | **2.67** | **2.22** | **3.29** |
| GID-shift | Ours | **310** | | GID-shift | Ours | 0.30 | 0.53 | 0.70 | 0.95 |
|  | H2 | - | |  | H2 | 0.18 | 0.21 | 0.28 | 0.41 |
|  | H5 | - | |  | H5 | 0.36 | 0.56 | 0.53 | 0.81 |
|  | Wald | - | |  | Wald | 0.77 | 0.58 | 0.34 | 0.23 |
|  | DK | - | |  | DK | **1.91** | **2.67** | **2.37** | **3.42** |

## B  ADDITIONAL BASELINE

We include one additional baseline, adapted version of Deep Kernel MMD (Liu et al., 2020). This Deep Kernel (DK) requires training of the kernel parameters and the network for extracting features. We use half of samples for this training process and conduct a test using the rest of samples as H2. We run this algorithm for Natural shift as in Section 5.2.

**Results.** The experimental results with DK are shown in Figure 4, and Table 4. As described in Section 5.2, Figure 4 shows the detection rates of each algorithms, and Table 4 presents FPR at selected points. Our approach outperforms this adapted version of Deep kernel MMD approach. First, in terms of the false positive rate (FPR), DK violates the desired FPR bound, while ours satisfies the bound (Table 4b). Furthermore, Table 4a shows that our approach generally requires a smaller number of samples for detection. In the M-shift case, ours needs ten more samples. In the other two shifts, however, ours shows superior performance compared to DK. In the GI-shift, our approach requires 150 fewer samples, and in GID-shift, our approach uses 310 samples for the detection, while DK cannot achieve 80 % detection.

## C  ADDITIONAL DISCUSSION

### C.1  MULTIPLE EPOCHS IN TRAINING

As shown in Algorithm 1, each example is used only once in updating the source-target classifier, baselines also follow this setting in all experiments. We consider this single epoch update anticipating that our algorithms being used in the online setting, where it is infeasible to take multiple passes

over the training data. However, without consideration of the online setting, each example can be used multiple times during training with the restriction that the example can be used only once in the CP interval, which can improve the performance. As this strategy is orthogonal to our approach, it can be applied to both ours and other baselines expecting the performance improvement.

# D  ADDITIONAL EXPERIMENTAL RESULTS

## D.1  DETECTION RATE

This section shows the additional detection rate plots for the different perturbations, severities, and window sizes ($w$) including figures in the main paper.

### D.1.1  M-SHIFT

Figure 5 - Figure 19 display the detection rate plot for M-shift scenario with different settings. These all different settings show the similar pattern with the figures in the main paper.

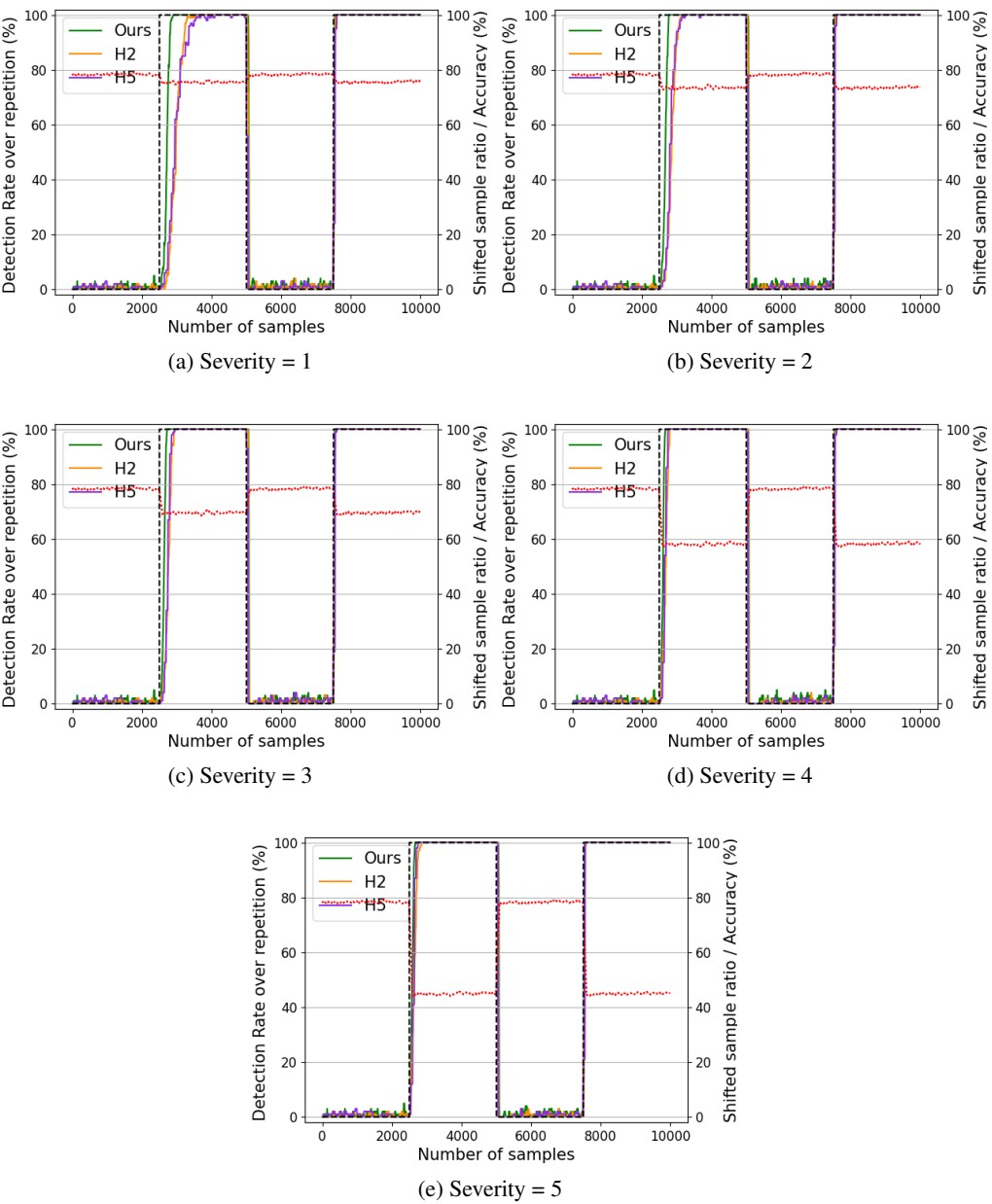

(a) Severity = 1

(b) Severity = 2

(c) Severity = 3

(d) Severity = 4

(e) Severity = 5

Figure 5: Contrast with $R = 100$, $w = 10$, $\alpha = 1\%$.

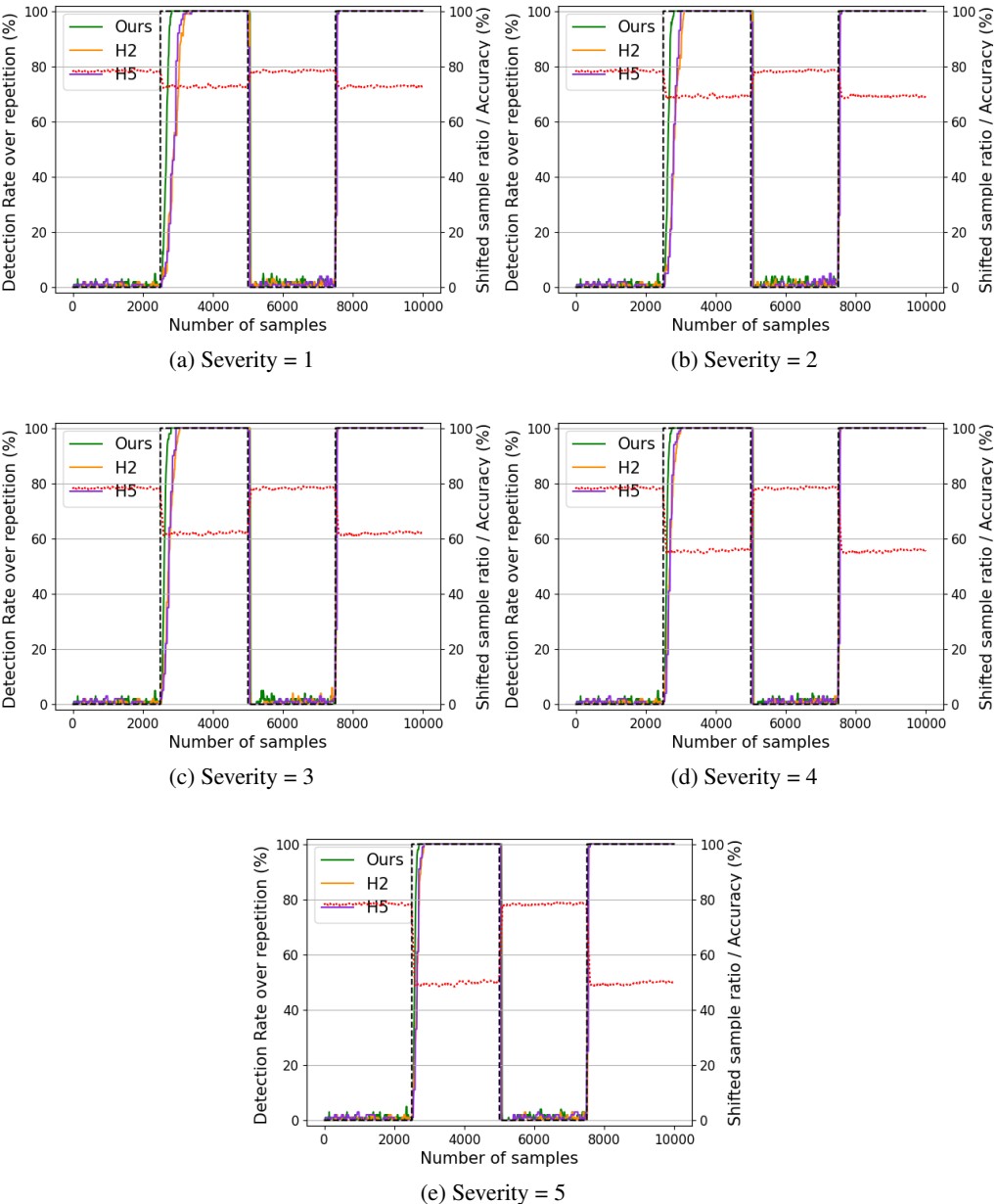

Figure 6: Defocus blur with $R = 100$, $w = 10$, $\alpha = 1\%$.

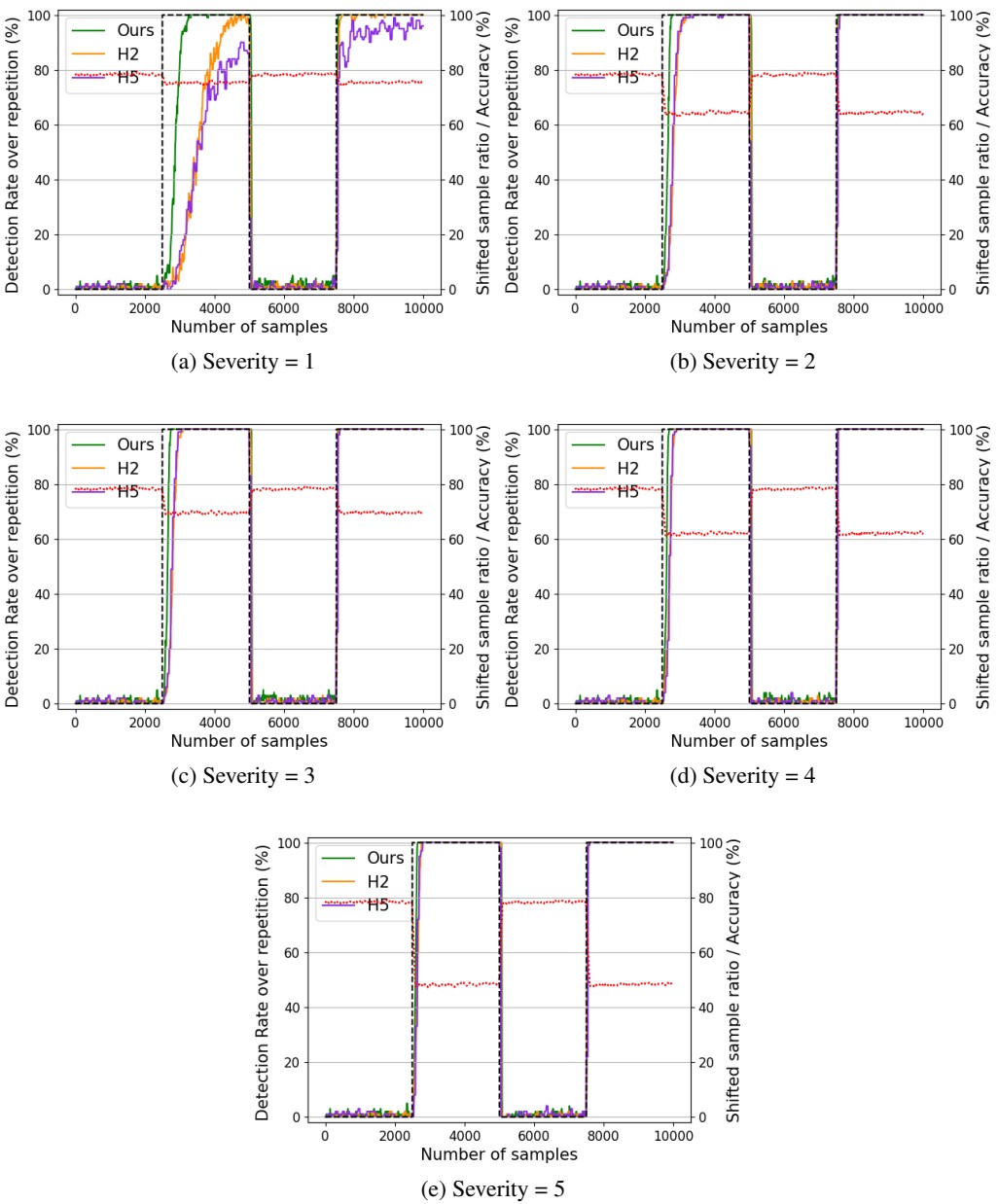

Figure 7: Elastic transform with $R = 100$, $w = 10$, $\alpha = 1\%$.

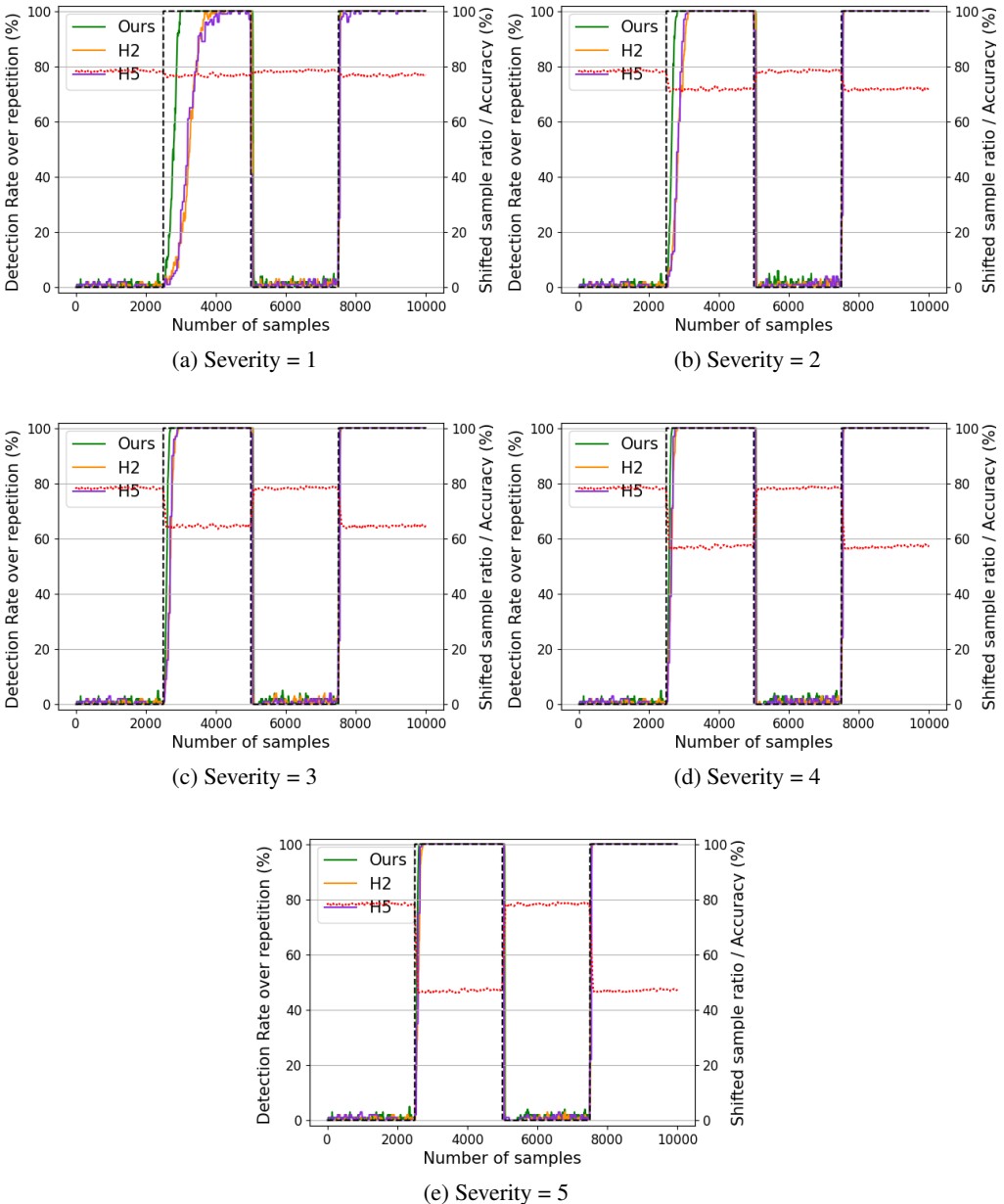

Figure 8: Gaussian blur with $R = 100$, $w = 10$, $\alpha = 1\%$.

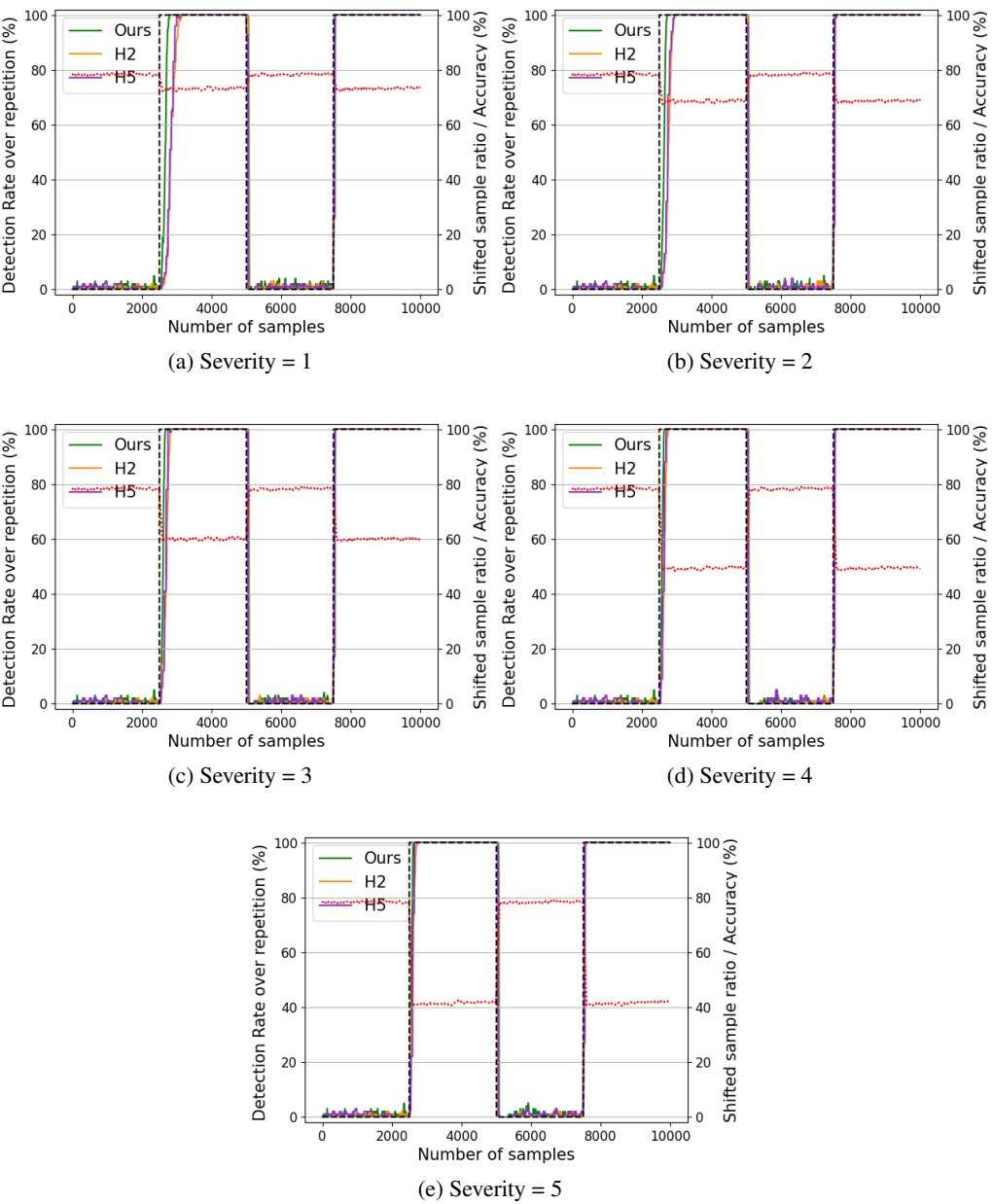

(a) Severity = 1

(b) Severity = 2

(c) Severity = 3

(d) Severity = 4

(e) Severity = 5

Figure 9: Gaussian noise with $R = 100$, $w = 10$, $\alpha = 1\%$.

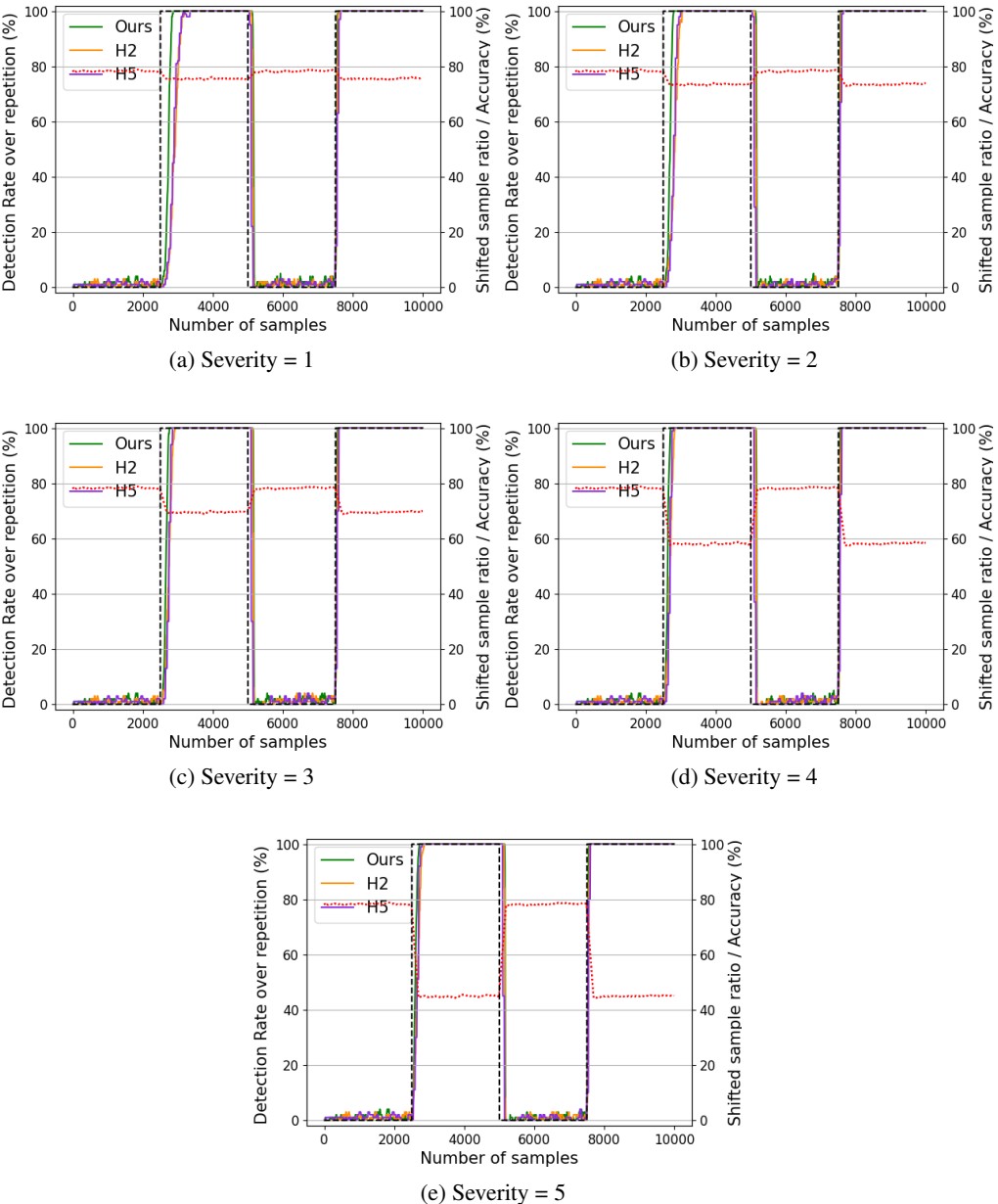

Figure 10: Contrast with $R = 100$, $w = 20$, $\alpha = 1\%$.

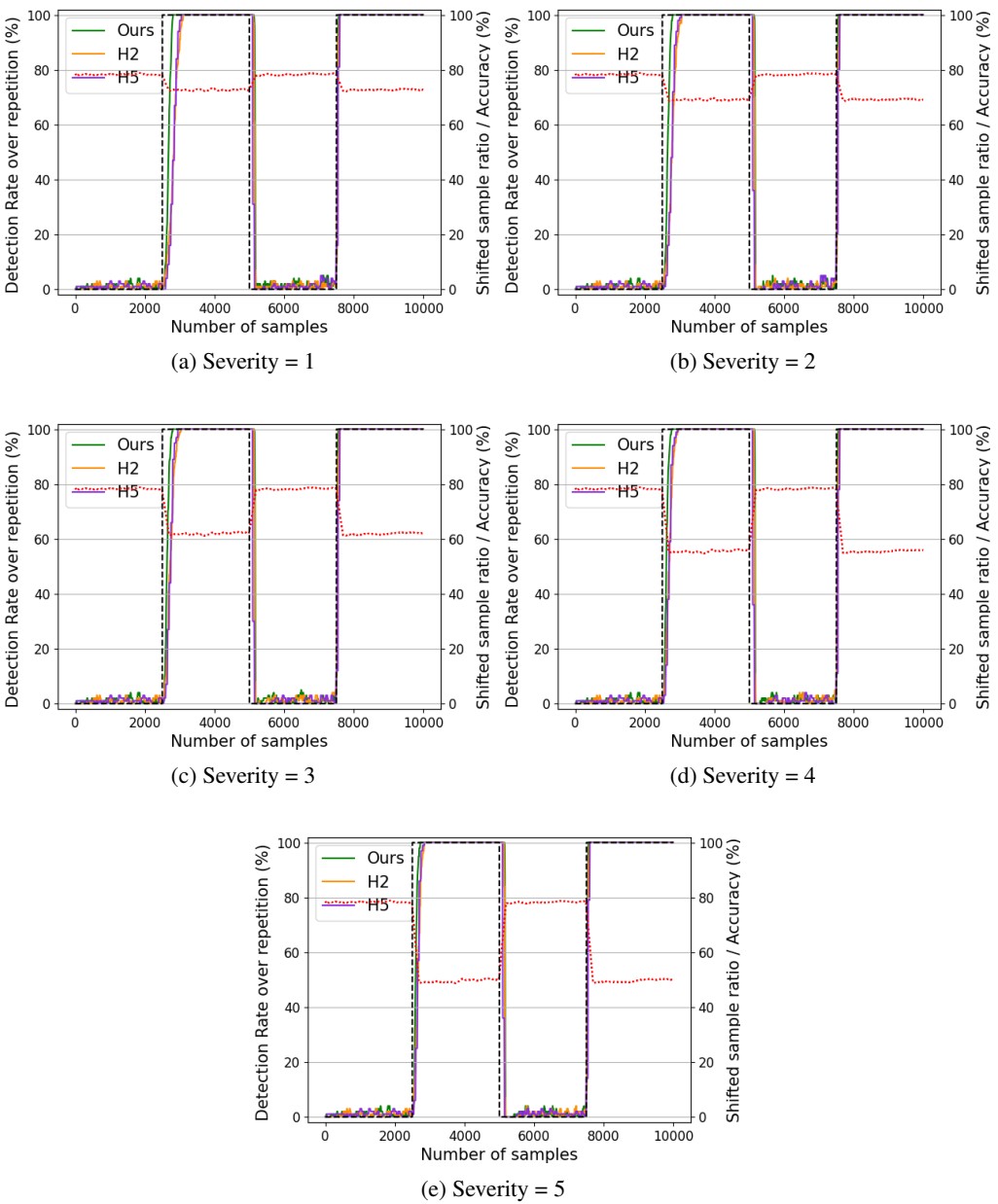

(a) Severity = 1

(b) Severity = 2

(c) Severity = 3

(d) Severity = 4

(e) Severity = 5

Figure 11: Defocus blur with $R = 100$, $w = 20$, $\alpha = 1\%$.

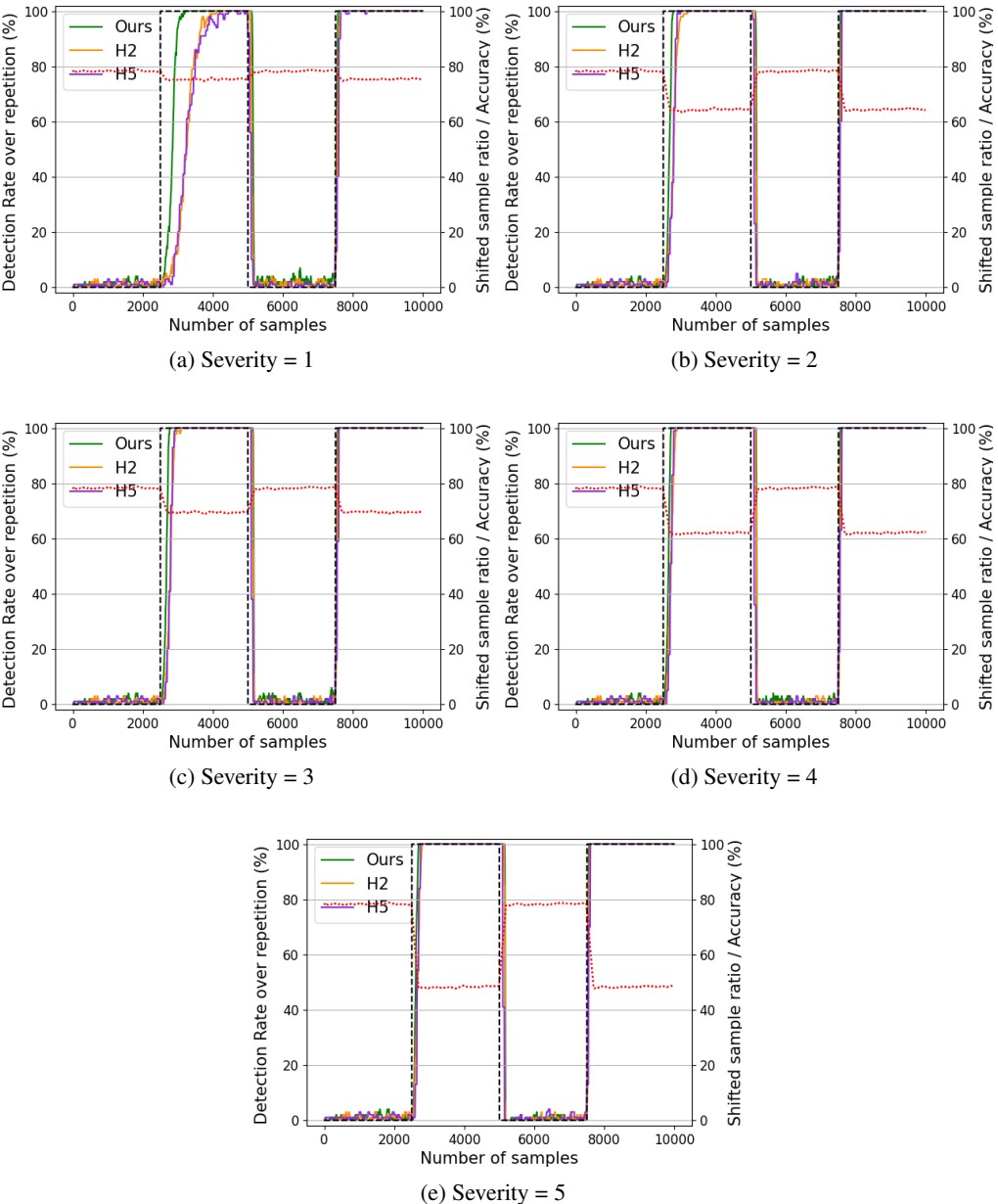

Figure 12: Elastic transform with $R = 100, w = 20, \alpha = 1\%$.

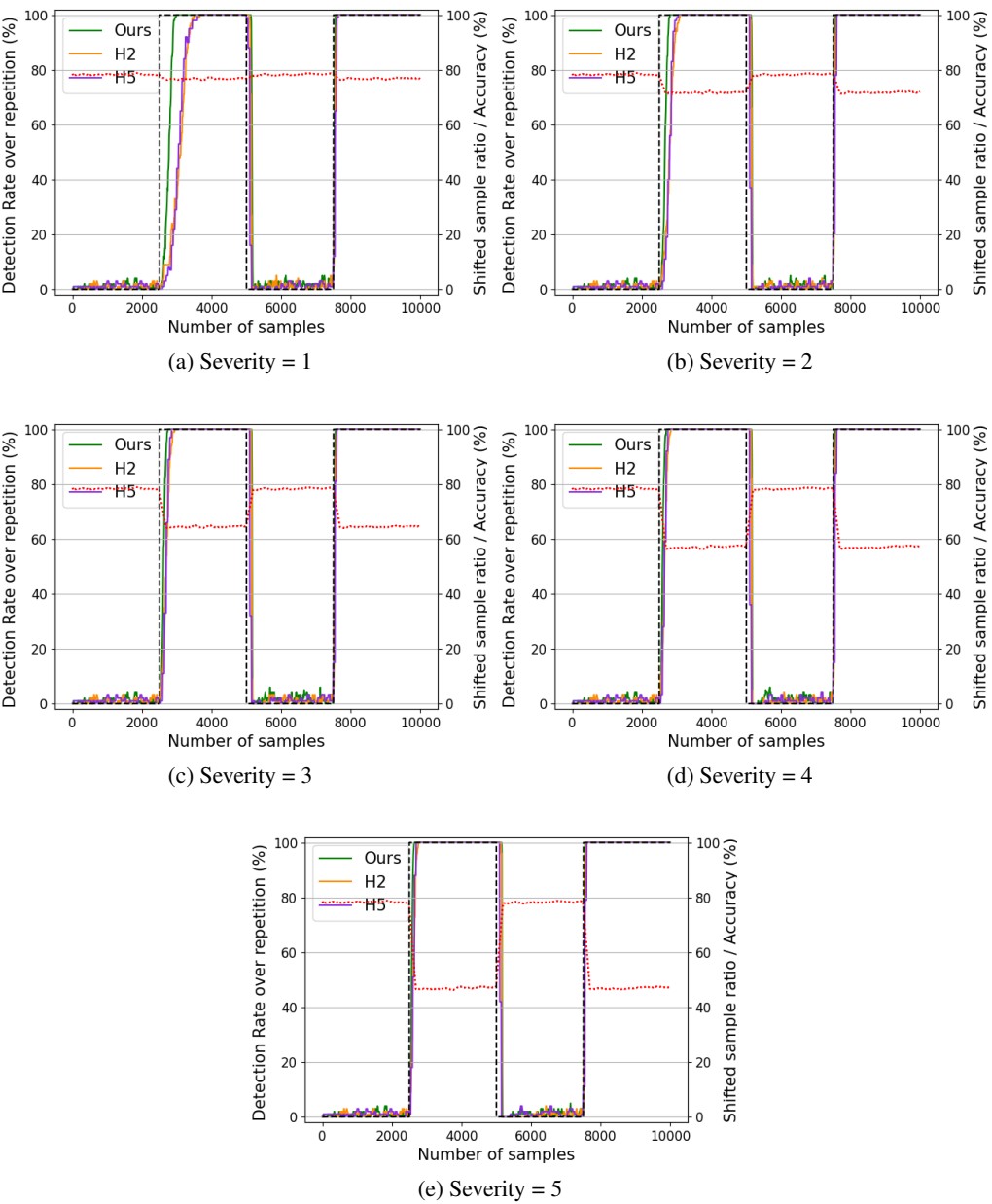

(a) Severity = 1

(b) Severity = 2

(c) Severity = 3

(d) Severity = 4

(e) Severity = 5

Figure 13: Gaussian blur with $R = 100$, $w = 20$, $\alpha = 1\%$.

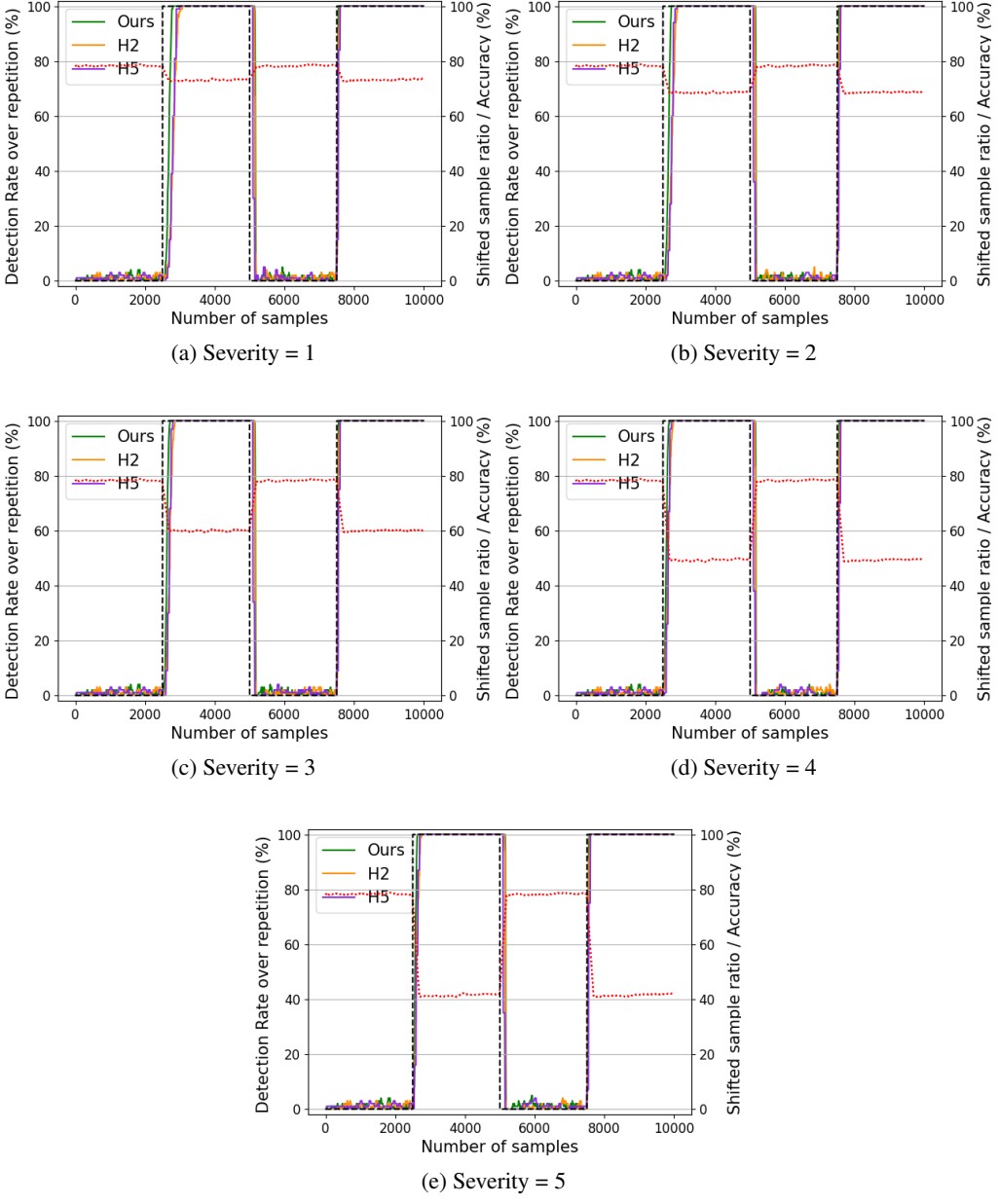

(a) Severity = 1

(b) Severity = 2

(c) Severity = 3

(d) Severity = 4

(e) Severity = 5

Figure 14: Gaussian noise with $R = 100$, $w = 20$, $\alpha = 1\%$.

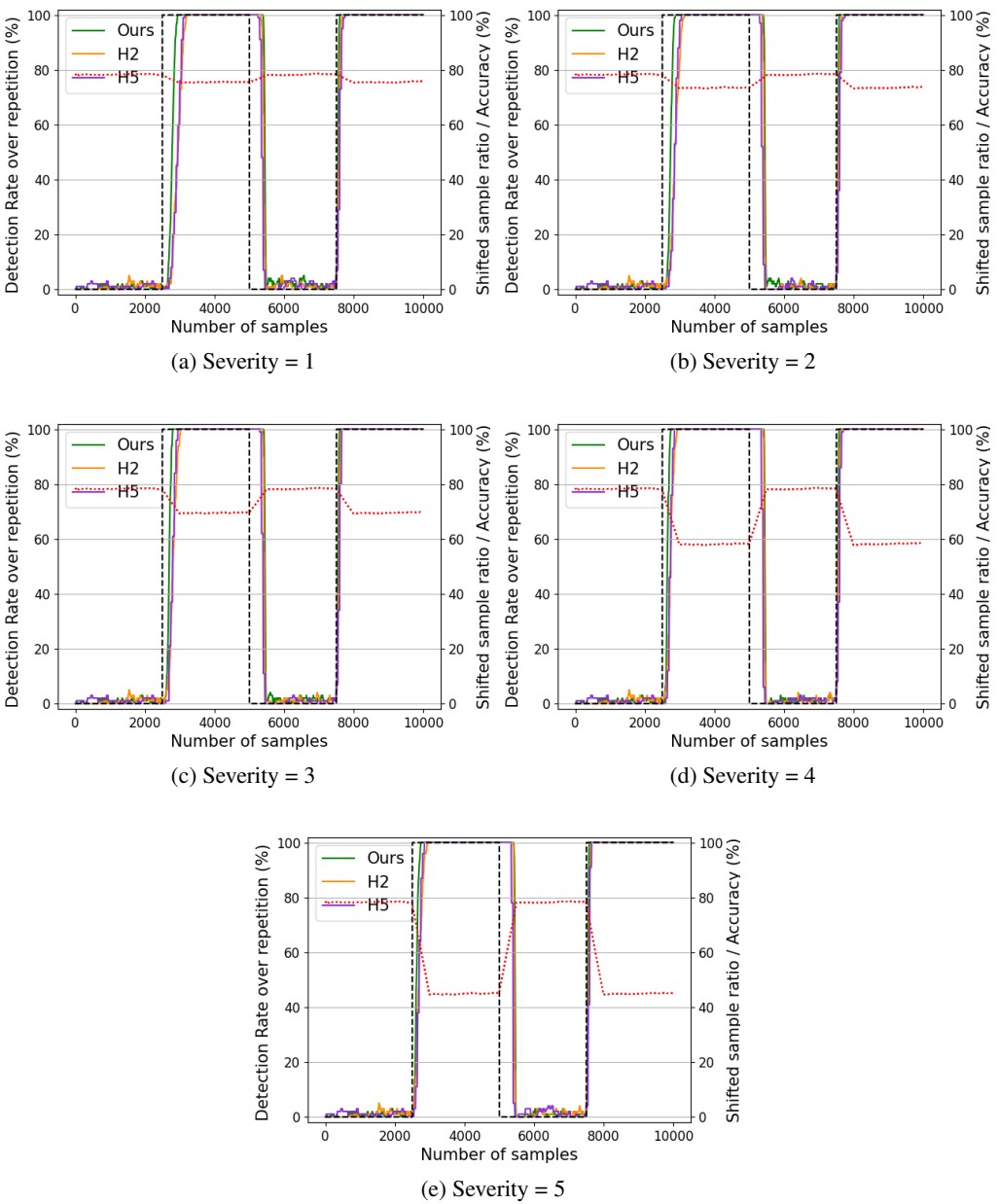

(a) Severity = 1

(b) Severity = 2

(c) Severity = 3

(d) Severity = 4

(e) Severity = 5

Figure 15: Contrast with $R = 100$, $w = 50$, $\alpha = 1\%$.

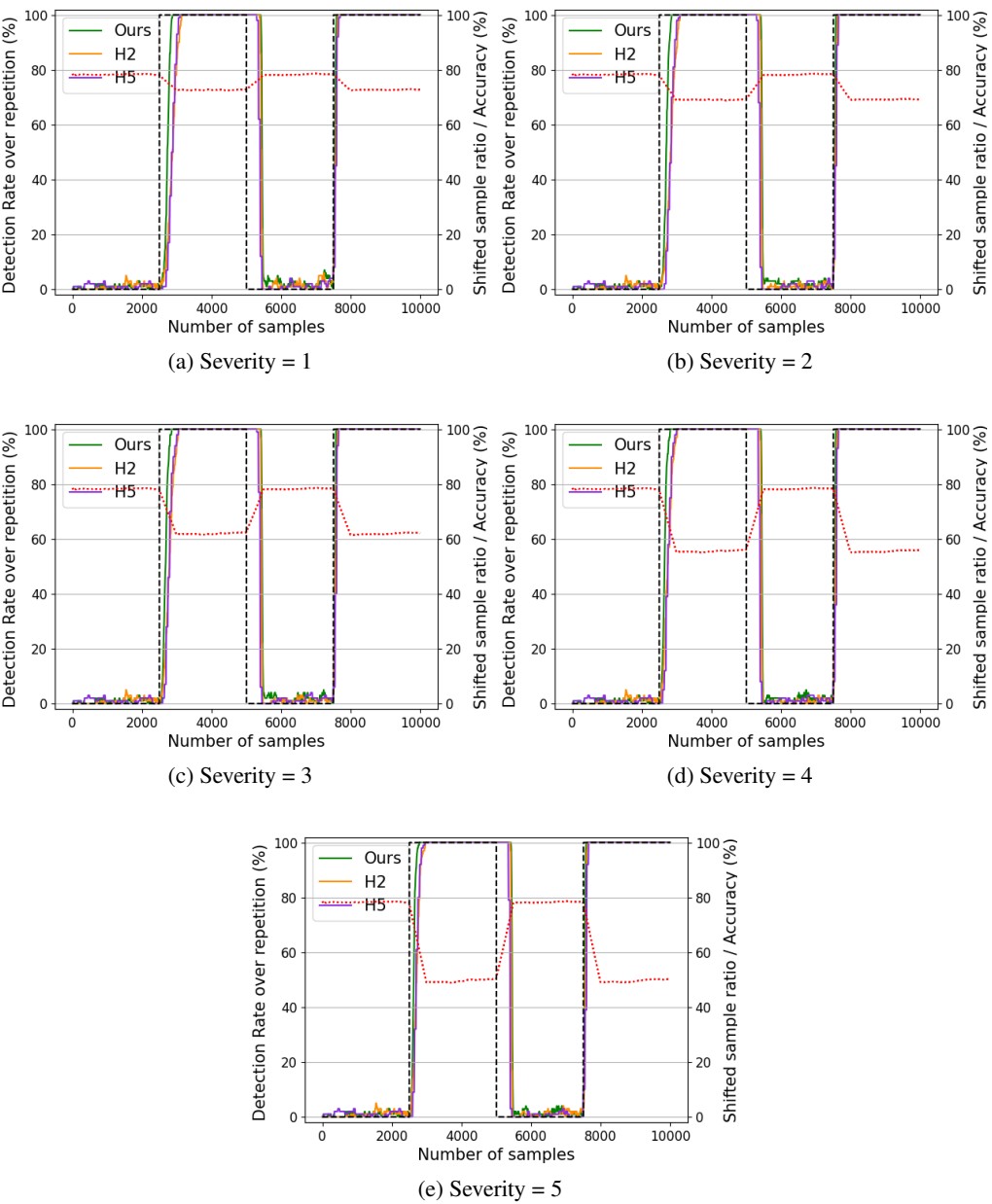

Figure 16: Defocus blur with $R = 100$, $w = 50$, $\alpha = 1\%$.

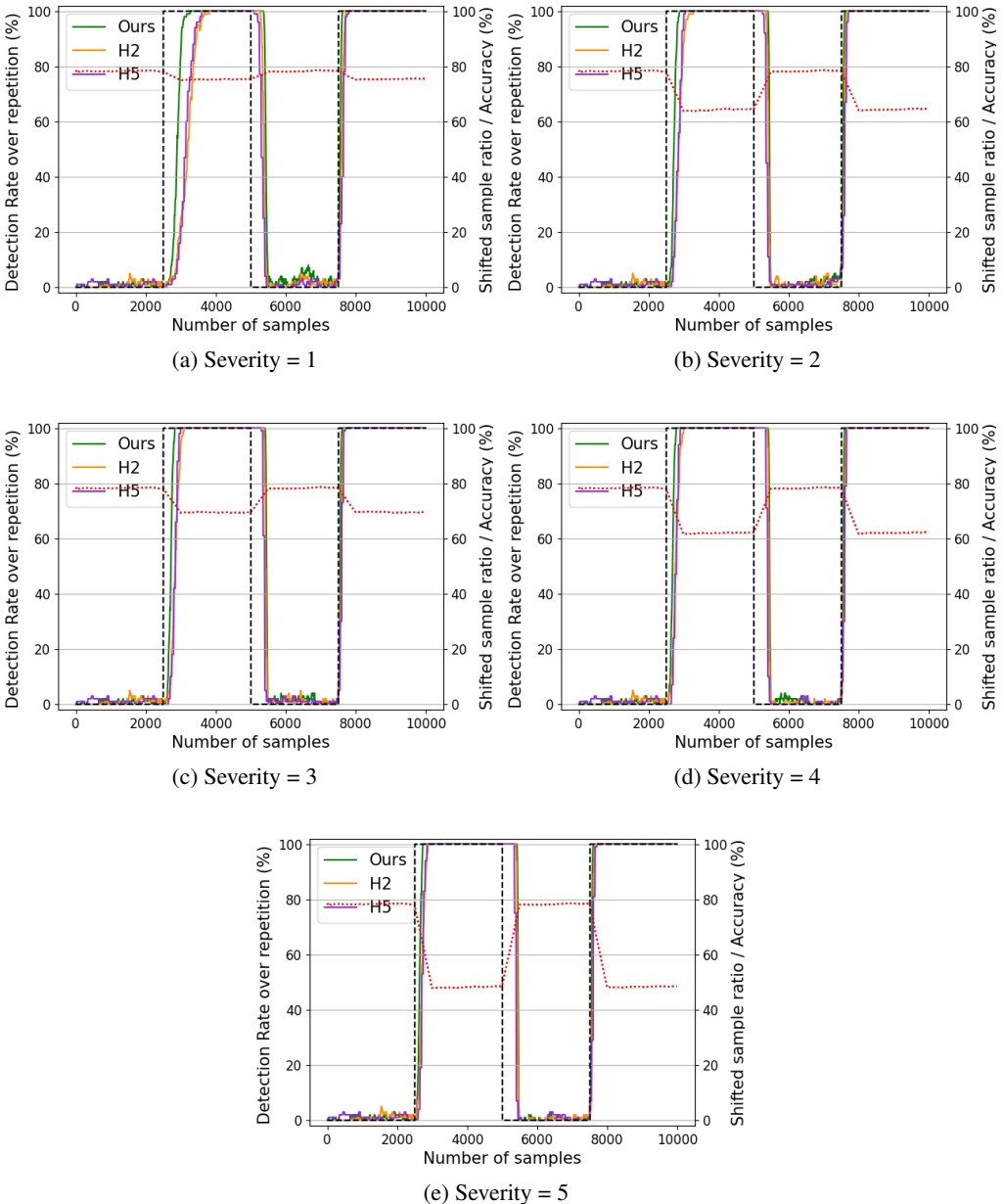

(a) Severity = 1

(b) Severity = 2

(c) Severity = 3

(d) Severity = 4

(e) Severity = 5

Figure 17: Elastic transform with $R = 100, w = 50, \alpha = 1\%$.

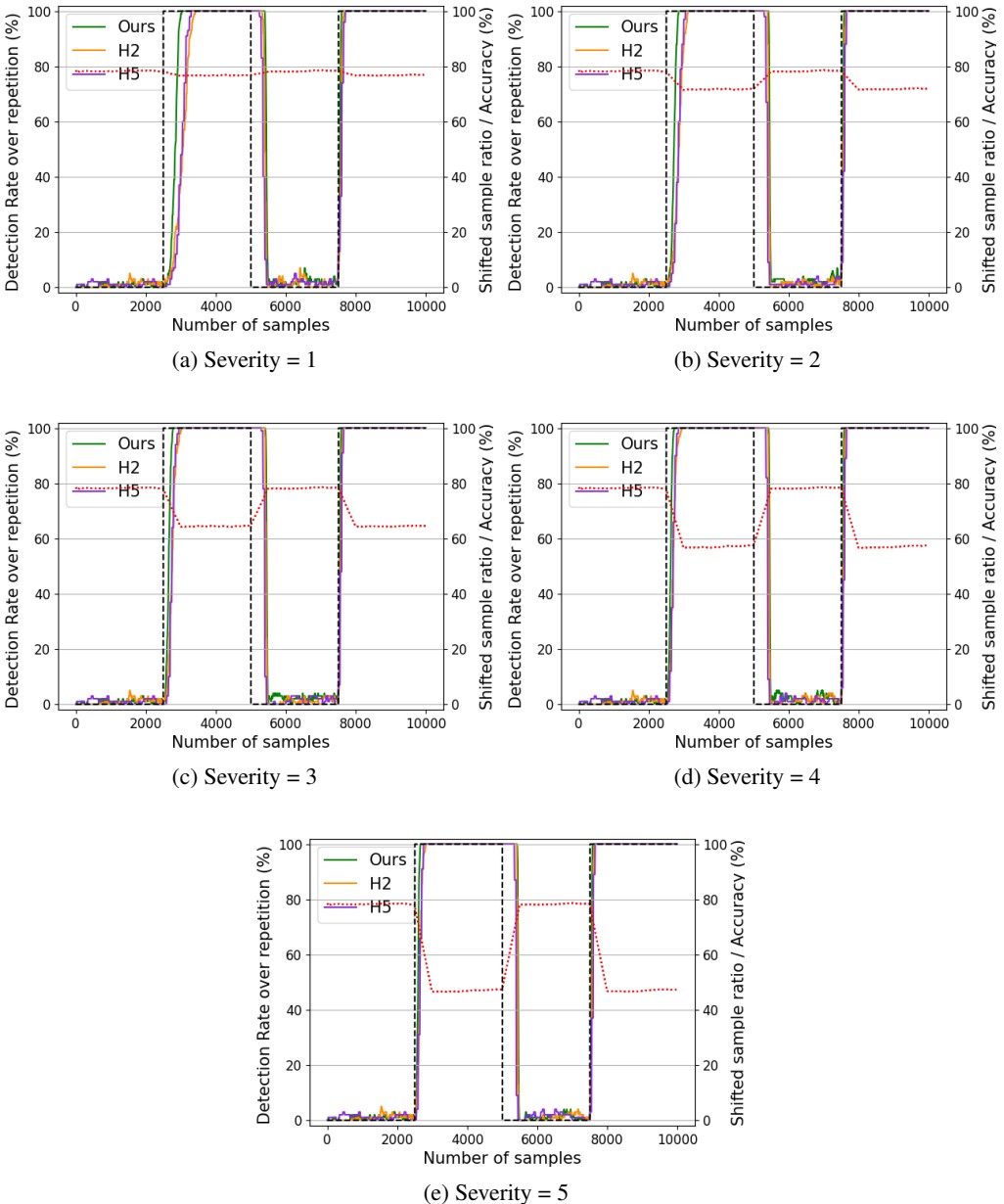

(a) Severity = 1

(b) Severity = 2

(c) Severity = 3

(d) Severity = 4

(e) Severity = 5

Figure 18: Gaussian blur with $R = 100$, $w = 50$, $\alpha = 1\%$.

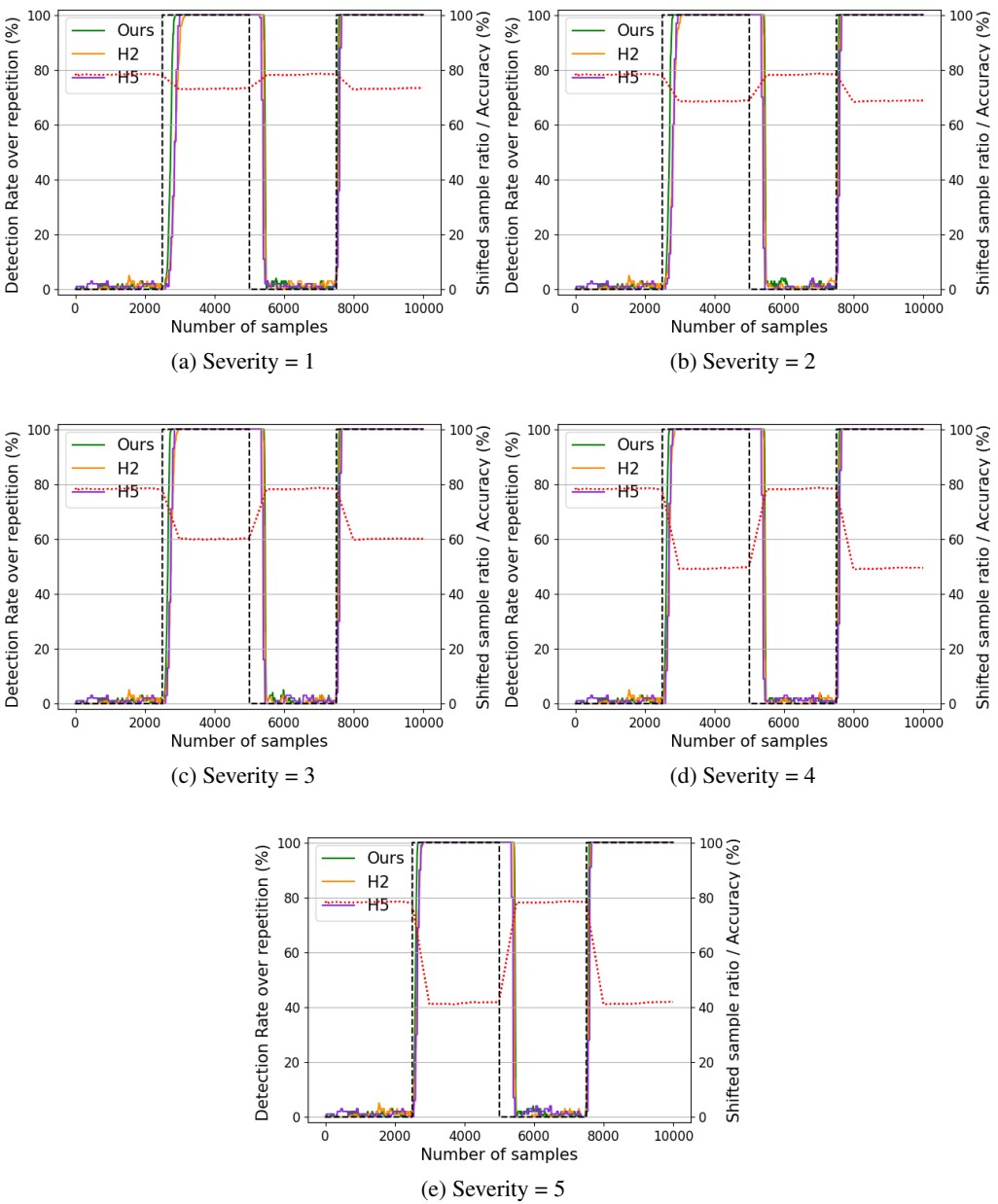

Figure 19: Gaussian noise with $R = 100$, $w = 50$, $\alpha = 1\%$.

### D.1.2    GI-SHIFT

This section includes the plots for GI-shift scenario with different perturbation, window sizes ($w$), and fixed severity.

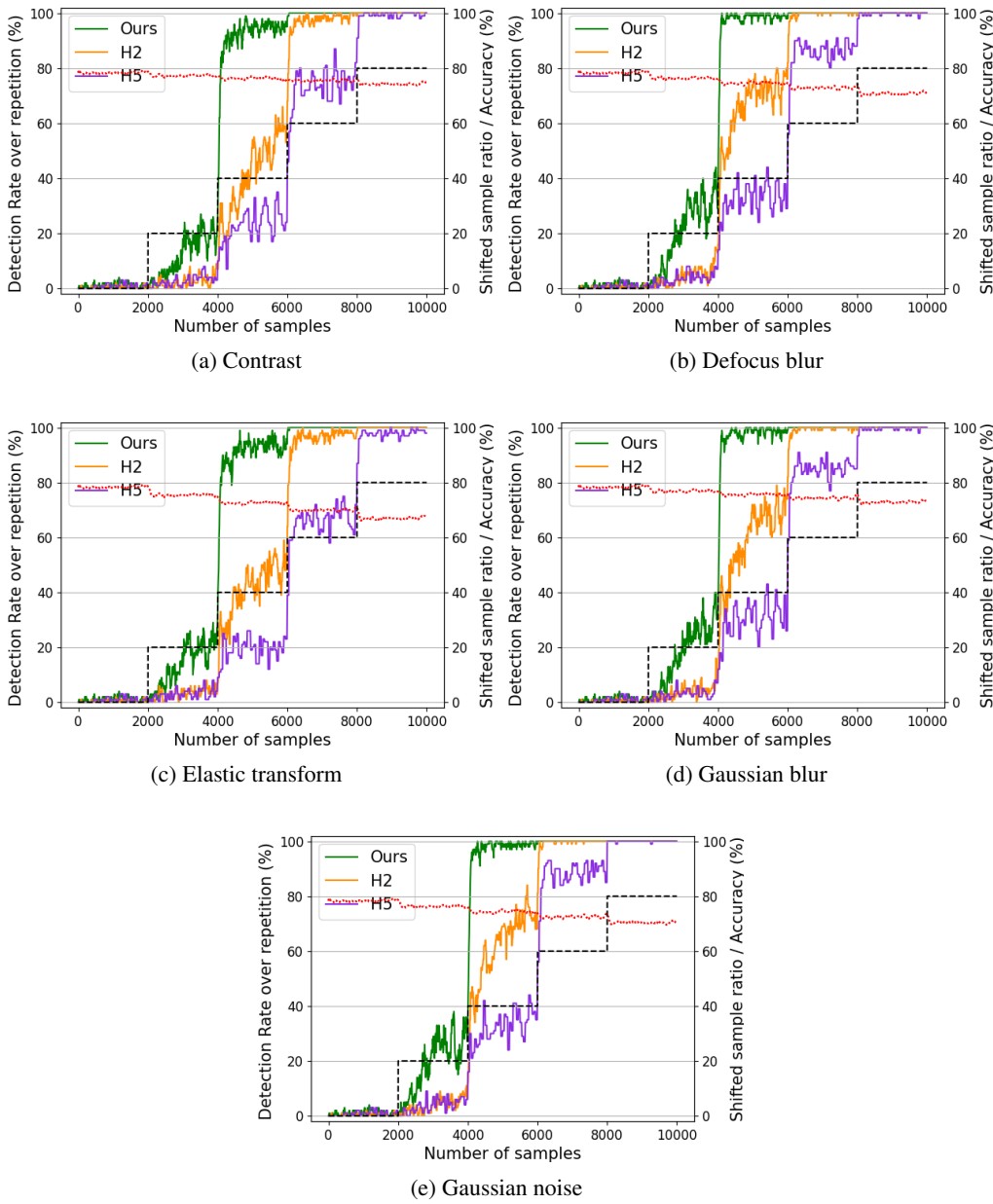

Figure 20: GI-shift with $R = 100$, $w = 10$, $\alpha = 1\%$.

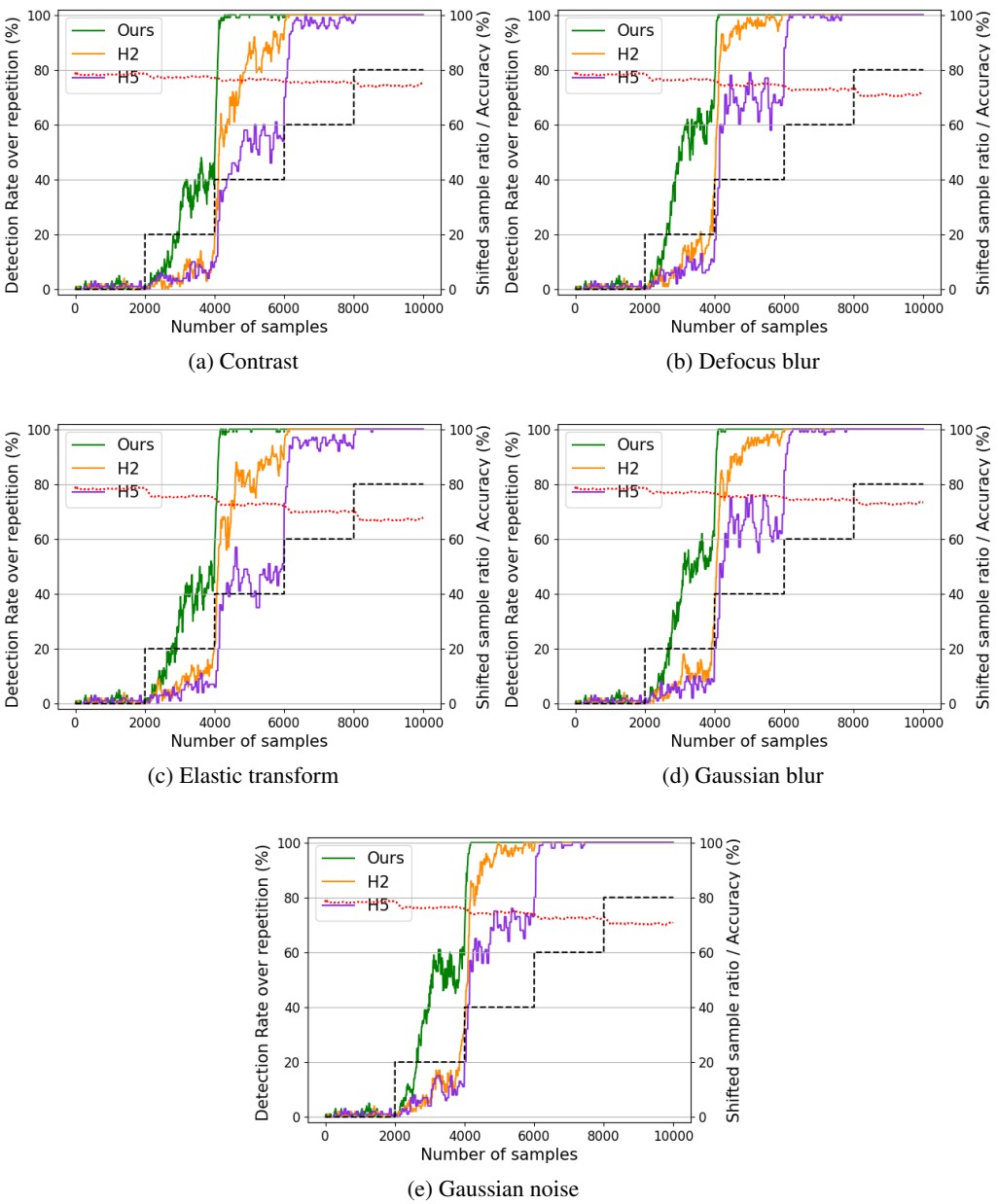

Figure 21: GI-shift with $R = 100$, $w = 20$, $\alpha = 1\%$.

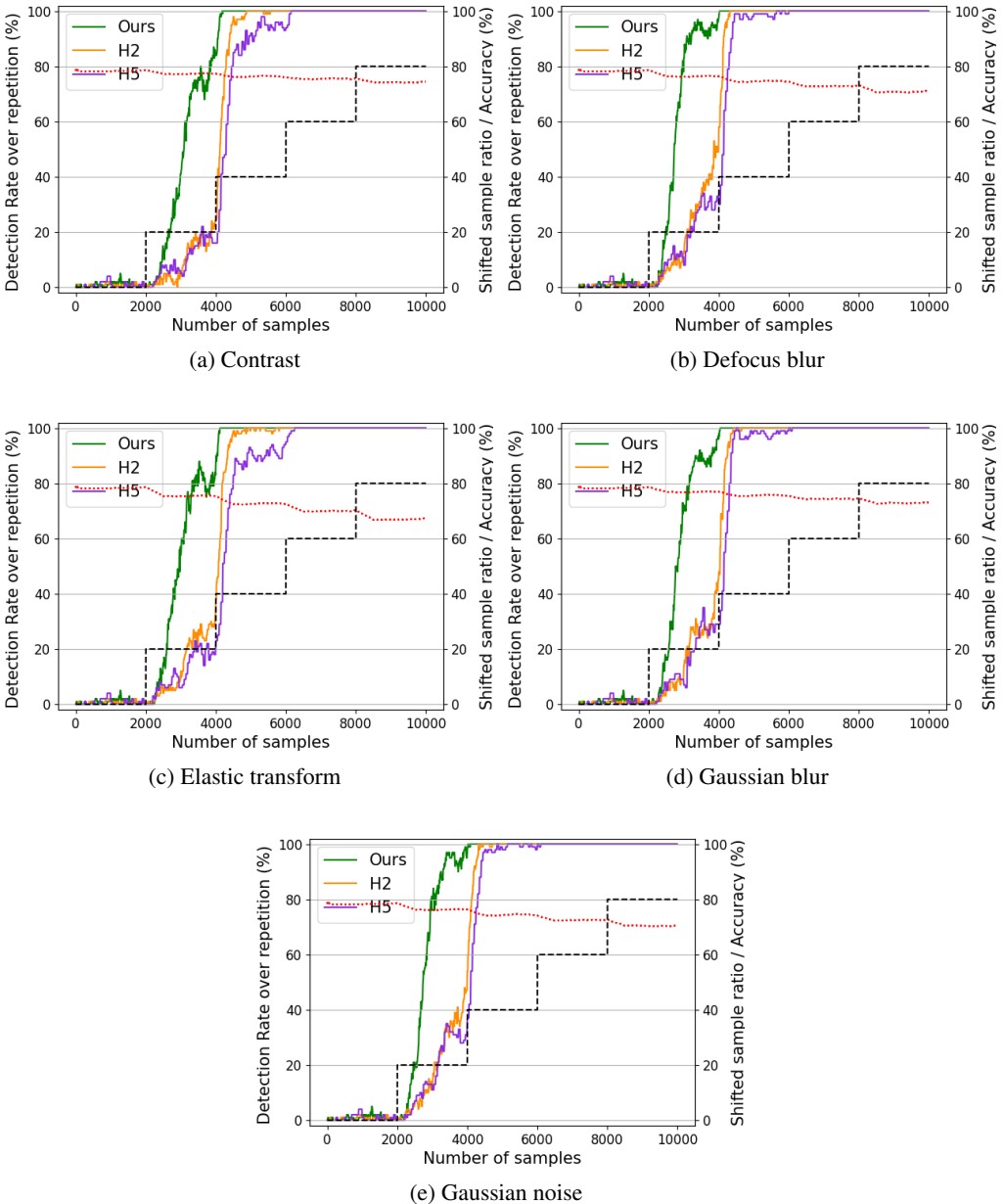

Figure 22: GI-shift with $R = 100$, $w = 50$, $\alpha = 1\%$.

### D.1.3 GID-SHIFT

Similar to the previous two sections, this section includes figures for the GID-shift scenario with different perturbation with severity 2, and different window sizes ($w$).

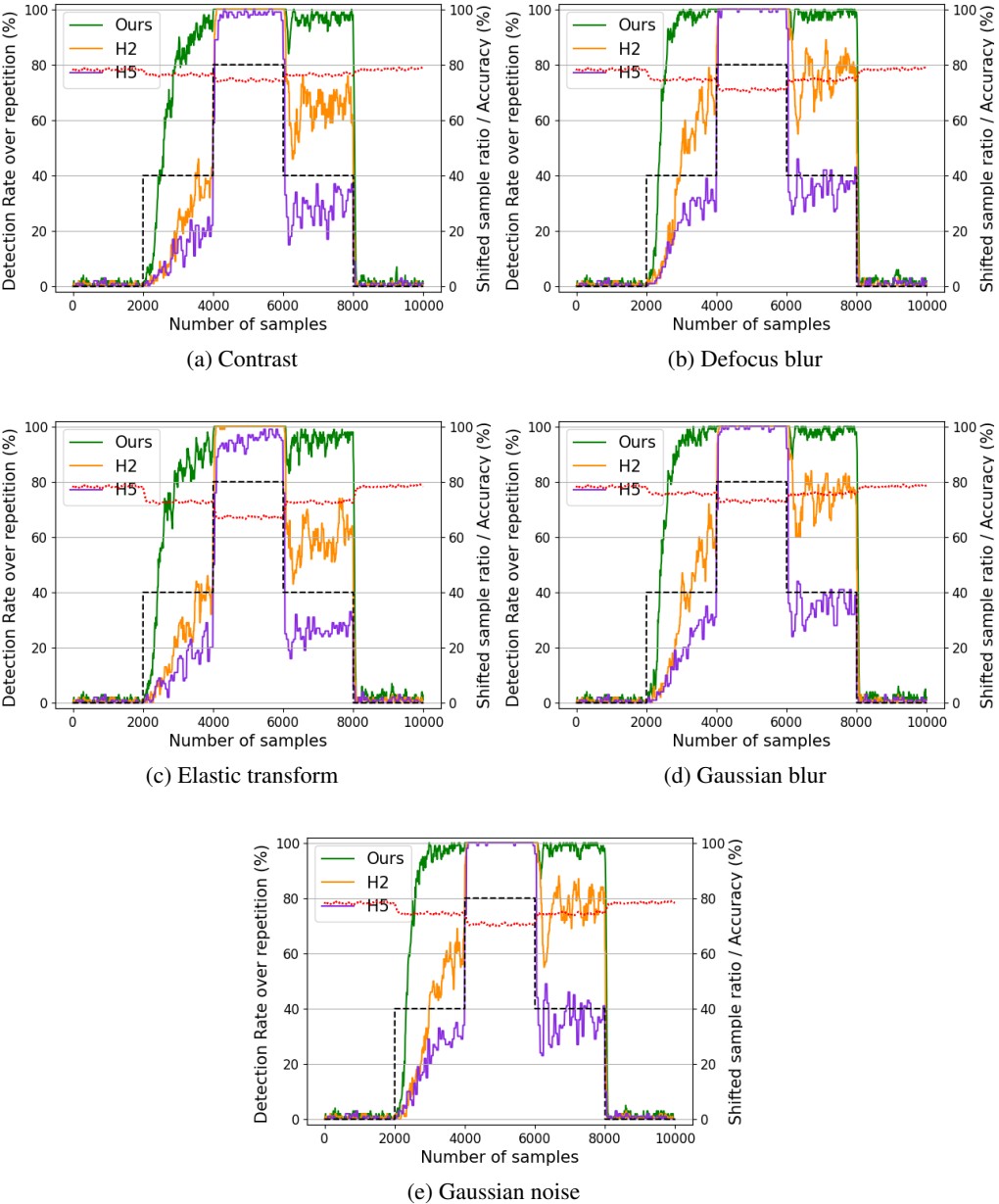

Figure 23: GID-shift with $R = 100$, $w = 10$, $\alpha = 1\%$.

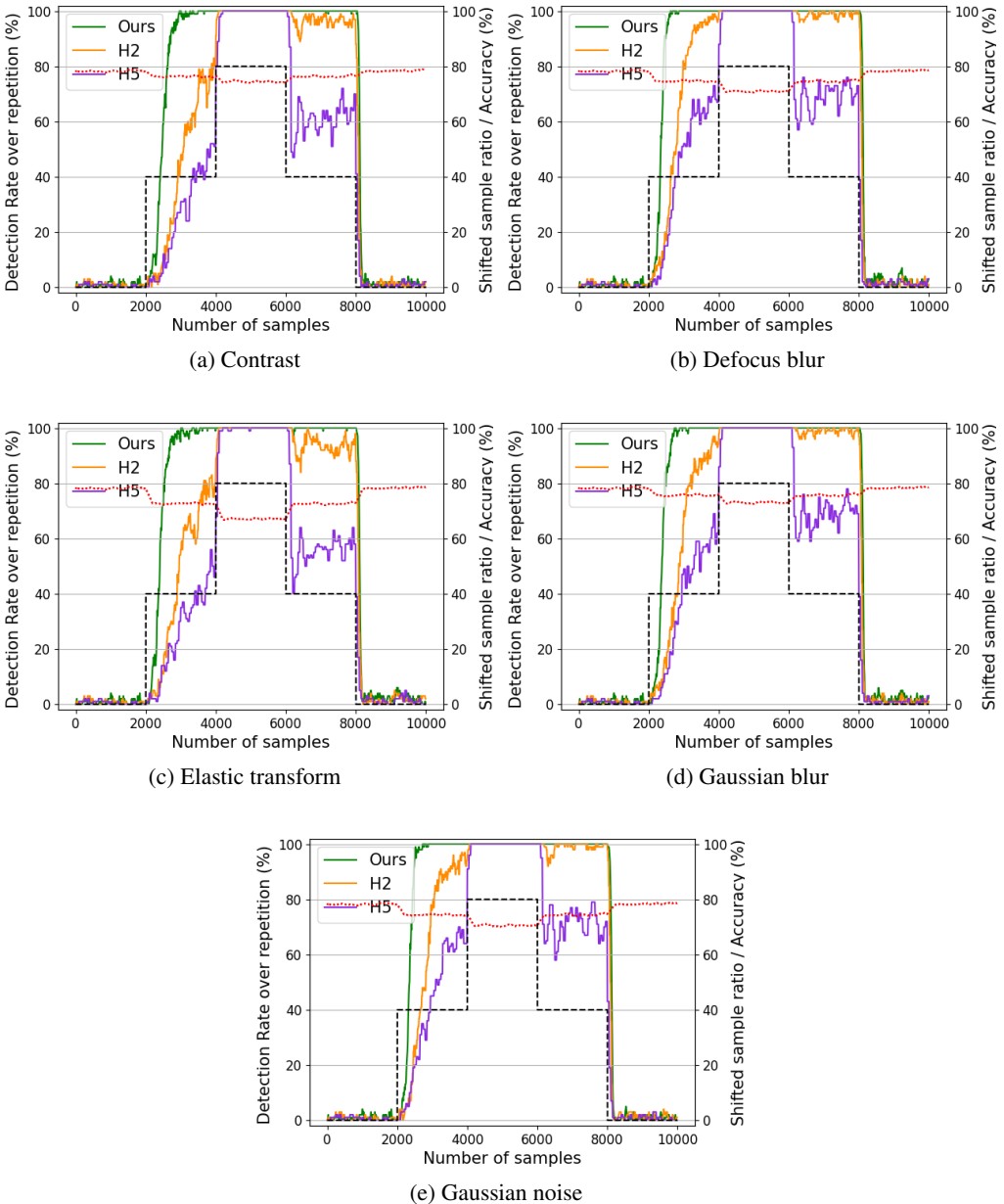

Figure 24: GID-shift with $R = 100$, $w = 20$, $\alpha = 1\%$.

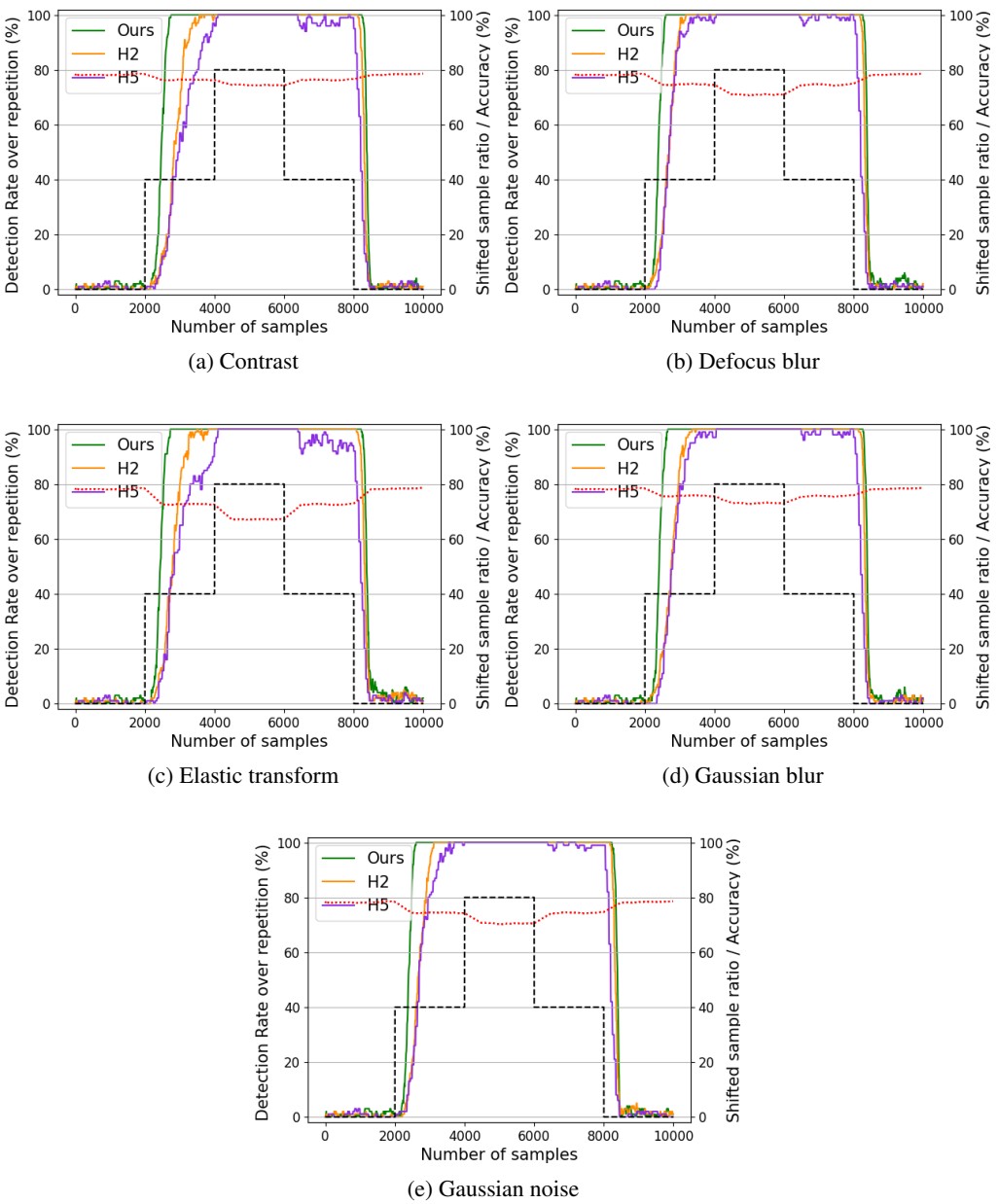

Figure 25: GID-shift with $R = 100$, $w = 50$, $\alpha = 1\%$.

### D.2 NUMBER OF SAMPLES FOR DETECTION

This section presents the required number of samples for detecting covariate shift over repetitions (Rate $\geq 80\%$) with different perturbations, severities, and window sizes ($w$).

### D.2.1 M-SHIFT

Table 5: Number of samples for detection with $R = 100$, $w = 10$

| Severity | Algorithms | Contrast | Defocus Blur | Elastic Transform | Gaussian Blur | Gaussian Noise |
|---|---|---|---|---|---|---|
| 1 | Ours | **280** | **240** | **510** | **400** | **220** |
|   | H2 | 610 | 550 | 1310 | 950 | 450 |
|   | H5 | 610 | 460 | 1610 | 960 | 410 |
| 2 | Ours | **230** | **200** | **220** | **210** | **180** |
|   | H2 | 470 | 450 | 450 | 490 | 350 |
|   | H5 | 410 | 410 | 410 | 460 | 310 |
| 3 | Ours | **190** | **170** | **190** | **140** | **140** |
|   | H2 | 350 | 370 | 410 | 270 | 270 |
|   | H5 | 310 | 360 | 360 | 260 | 260 |
| 4 | Ours | **150** | **140** | **160** | **120** | **120** |
|   | H2 | 250 | 290 | 290 | 210 | 210 |
|   | H5 | 210 | 260 | 260 | 210 | 160 |
| 5 | Ours | **130** | **120** | **140** | **100** | **110** |
|   | H2 | 210 | 230 | 210 | 170 | 150 |
|   | H5 | 160 | 210 | 210 | 160 | 160 |

Table 6: Number of samples for detection with $R = 100$, $w = 20$

| Severity | Algorithms | Contrast | Defocus Blur | Elastic Transform | Gaussian Blur | Gaussian Noise |
|---|---|---|---|---|---|---|
| 1 | Ours | **300** | **250** | **430** | **360** | **250** |
|   | H2 | 530 | 430 | 970 | 770 | 410 |
|   | H5 | 460 | 410 | 1010 | 710 | 360 |
| 2 | Ours | **250** | **220** | **240** | **230** | **210** |
|   | H2 | 430 | 390 | 390 | 410 | 350 |
|   | H5 | 360 | 360 | 360 | 360 | 310 |
| 3 | Ours | **220** | **200** | **230** | **170** | **170** |
|   | H2 | 330 | 330 | 370 | 290 | 270 |
|   | H5 | 310 | 310 | 360 | 260 | 260 |
| 4 | Ours | **170** | **170** | **200** | **140** | **140** |
|   | H2 | 270 | 290 | 310 | 230 | 230 |
|   | H5 | 260 | 260 | 260 | 210 | 210 |
| 5 | Ours | **160** | **150** | **170** | **110** | **120** |
|   | H2 | 250 | 250 | 230 | 190 | 190 |
|   | H5 | 210 | 210 | 210 | 160 | 160 |

Table 7: Number of samples for detection with $R = 100$, $w = 50$

| Severity | Algorithms | Contrast | Defocus Blur | Elastic Transform | Gaussian Blur | Gaussian Noise |
|---|---|---|---|---|---|---|
| 1 | Ours | **360** | **300** | **490** | **430** | **290** |
| | H2 | 570 | 510 | 870 | 710 | 470 |
| | H5 | 560 | 460 | 810 | 660 | 410 |
| 2 | Ours | **310** | **270** | **280** | **280** | **250** |
| | H2 | 510 | 450 | 470 | 470 | 390 |
| | H5 | 460 | 410 | 460 | 460 | 360 |
| 3 | Ours | **260** | **240** | **270** | **220** | **210** |
| | H2 | 410 | 410 | 450 | 330 | 330 |
| | H5 | 360 | 360 | 410 | 310 | 310 |
| 4 | Ours | **210** | **210** | **230** | **170** | **160** |
| | H2 | 310 | 370 | 370 | 270 | 270 |
| | H5 | 310 | 360 | 360 | 260 | 260 |
| 5 | Ours | **170** | **170** | **190** | **130** | **130** |
| | H2 | 290 | 310 | 290 | 210 | 210 |
| | H5 | 260 | 260 | 260 | 210 | 210 |

### D.2.2 GI-SHIFT

Table 8: Number of samples for detection with $R = 100, w = 20$

| Algorithms | Contrast | Defocus Blur | Elastic Transform | Gaussian Blur | Gaussian Noise |
|---|---|---|---|---|---|
| Ours | **2100** | **2040** | **2100** | **2040** | **2050** |
| H2 | 2890 | 2150 | 2610 | 2170 | 2190 |
| H5 | 4110 | 4010 | 4110 | 4010 | 4010 |

Table 9: Number of samples for detection with $R = 100, w = 50$

| Algorithms | Contrast | Defocus Blur | Elastic Transform | Gaussian Blur | Gaussian Noise |
|---|---|---|---|---|---|
| Ours | **1580** | **980** | **1340** | **1170** | **970** |
| H2 | 2310 | 2130 | 2190 | 2170 | 2130 |
| H5 | 2460 | 2310 | 2510 | 2310 | 2310 |

### D.2.3 GID-SHIFT

Table 10: Number of samples for detection with $R = 100, w = 20$

| Algorithms | Contrast | Defocus Blur | Elastic Transform | Gaussian Blur | Gaussian Noise |
|---|---|---|---|---|---|
| Ours | **620** | **430** | **530** | **470** | **450** |
| H2 | 1890 | 990 | 1670 | 1170 | 1030 |
| H5 | 2010 | 2010 | 2060 | 2010 | 2010 |

Table 11: Number of samples for detection with $R = 100, w = 50$

| Algorithms | Contrast | Defocus Blur | Elastic Transform | Gaussian Blur | Gaussian Noise |
|---|---|---|---|---|---|
| Ours | **590** | **480** | **550** | **520** | **490** |
| H2 | 1110 | 870 | 1010 | 970 | 910 |
| H5 | 1510 | 910 | 1310 | 1010 | 960 |

