# OpenReview forum: "Sequential Covariate Shift Detection Using Classifier Two-Sample Tests"
_ICLR.cc/2022/Conference — ICLR 2022 Submitted_

### Official Review · Reviewer_Wc7g · 2021-10-26

**Correctness:** 4
**Technical Novelty And Significance:** 3
**Empirical Novelty And Significance:** 3
**Recommendation:** 6
**Confidence:** 3

**Main Review:**

### Pros:
1) This work introduces a novel method to train the classifier and detect covariate shifts in a sequential manner. And the online pipeline is interesting.
2) This work is well formulated with elaborate proofs and theoretically solid.

### Cons:
1) This method evaluates each example before taking a gradient step on that example. It means that every sample can be used only once for training the classifier, i.e., training for one epochs. It seems to be a great limitation of this method. With a held-out test set, this problem doesn’t exist. As a common sense, training for more epochs can improve the performance. If the baselines (H2, H5) are better trained, I think their performances would be much better.
2) Besides, the experiments seem to be a litte unfair. The proposed method and baselines (H2, H5) differ in both training sets and testing sets because of the different sampling strategies. The proposed method use the whole set for both training and testing while baselines sample some examples for testing and use the others for training.
3) I also concern about the problem of cold boot. At the begining of training, the classifier is with poor performance. how can we evaluate the samples in this stage?




**Summary Of The Paper:**

This paper proposes a new covariate shift detection method, which distinguishes whether training data and test data come from the same distribution. This method sequentially evaluates each example before taking a gradient step on that example, avoiding constructing a held-out test set. Experiments on ImageNet validate its effectiveness at detecting both natural and synthetic covariate shifts.

**Summary Of The Review:**

Though this paper is interesting and theoretically solid, the experimental parts do not convince me. Currently my rating is marginally below the acceptance threshold. I will raise my score if the concerns about the experiments are addressed.

### After rebuttal
I raise the score from 5 to 6 given that the experimental parts are improved and most concerns are solved. But I will not act as a champion for acceptance.

---

> ### Author Response · Authors · 2021-11-23
> **Response to comments by Reviewer Wc7g**
>
> Thank you for the valuable comments and concerns. The following includes our response; we have updated our paper accordingly and highlighted the updates in red.
>
> 1. This method evaluates each example before taking a gradient step on that example. It means that every sample can be used only once for training the classifier, i.e., training for one epochs. It seems to be a great limitation of this method. With a held-out test set, this problem doesn’t exist. As a common sense, training for more epochs can improve the performance. If the baselines (H2, H5) are better trained, I think their performances would be much better.
>
> * **Response:** We agree that each example can be used multiple times for the baselines (H2, H5) to train the classifier. However, this strategy can also be used with our approach. Instead, the restriction is that we can only use each once in the CP interval, which is the case for both our approach and the baselines. We focused on a single gradient step since we anticipate these algorithms being used in the online setting, where it is infeasible to take multiple passes over the training data (also, this strategy is orthogonal to our contributions). We have added a discussion to our paper (In Appendix C.1).
>
> 2. Besides, the experiments seem to be a litte unfair. The proposed method and baselines (H2, H5) differ in both training sets and testing sets because of the different sampling strategies. The proposed method use the whole set for both training and testing while baselines sample some examples for testing and use the others for training.
>
> * **Response:** The ability to use all the data is our key contribution, and our experiments are designed to highlight this advantage of our approach. Unlike the baselines, which correspond to the classical Classifier Two-Sample Tests (C2STs), our approach does not require a separate held-out set to construct Clopper-Pearson intervals, so it can use the entire set of examples. In particular, Lemma 1 guarantees it is valid to use the held-out set in this way.
>
> 3. I also concern about the problem of cold boot. At the begining of training, the classifier is with poor performance. how can we evaluate the samples in this stage?
>
> * **Response:** If the classifier performs poorly, our approach will conservatively report “no shift” (in particular, the Clopper-Pearson interval we construct will be large). Our experiments show that we can detect shifts with very few examples, which suggests that cold boot is not a major problem. In general, it is very challenging to detect shifts with too few examples, since a small number of examples might simply be outliers in the source distribution.

---

> > ### Comment · Reviewer_Wc7g · 2021-11-28
> > **Response to authors**
> >
> > Thanks for the response. Most of my concerns are addressed.  And additional explanations and experiments make this work more credible.  I’d like to update my rating from 5 to 6 but will not argue strongly for acceptance.

---

### Official Review · Reviewer_PsoX · 2021-10-28

**Correctness:** 3
**Technical Novelty And Significance:** 3
**Empirical Novelty And Significance:** Not applicable
**Recommendation:** 5
**Confidence:** 5

**Main Review:**

Strong points:
- the method is simple and allows to use any classification model that can be trained in an online manner, which makes it of practical interest
- the paper is clearly written
- there is strong theoretical grounding coming from classical statistics

Weak points:
- there is some obscurity in the proof of the crucial theorem 3, which could hide a flaw
- an empirical comparison with existing two-sample test would be interesting to better justify the need for this method and as an additional baseline
- in the experiments, the false positive rate is above the specified level:
  - the given reason (multiple hypothesis testing) is not convincing
  - it makes the detection power comparison less meaningful

Detailed review:

### COMPARISON TO SEQUENTIAL TWO-SAMPLE TESTS:
It is argued (sections 1 and 2) that sequential tests are not appropriate in the scenarios proposed since they assumed that all the data is shifted. This assumption is only relevant with respect to False Negative Rate guarantees (e.g. consistency). Nevertheless, guarantees on FPR are obtained under the assumption that the distributions are equal. Therefore, sequential two-sample test are applicable to the scenario of this paper.
Providing comparisons with some of the cited sequential two-sample test would definitely make the paper stronger, justifying better the need for the proposed method and its advantages. You could run a sequential two-sample test by drawing samples from the training set (without replacement---this requires the training set to be roughly as large as the testing horizon) and from the test set in the order they come, until a rejection occurs and then you could reset everything and start testing again.

### LEMMA 1:
The notation $x\_{i..j}$ should be introduced and $i,j$ should be specified.
Does $\hat{g}\_{i,j-1}$ refer to the classifier itself (in the case of a NN a set of weights and biases?) ?  Or is it the accuracy on the range $i..j-1$?
In the proof, $\hat{g}_{i}$ appears as a random variable, so I guess it represents all the variables defining the model.
This should be formalized.

### PROOF OF THEOREM 3
It is not clear how Lemma 1 is used:  more precisely, **why can you say that  $s_{w,t}$ is a binomial variable with parameter 1/2?** Before Lemma 1, you say:
*the key to have valid bounds is proving the independence on predictions $\hat{y}\_1, \dots, \hat{y}\_t$ (and $\hat{y}'\_1, \dots, \hat{y}'\_t$) to have a valid Clopper-Pearson interval, since they are seemingly
dependent through online learned classifier $\hat{g}_t$.
First, our key result shows that our estimate of the
accuracy of $\hat{g}\_t$ valid—i.e., the labels $\hat{y}\_{i:j}$ are conditionally independent.*

What do you mean exactly by a valid estimate of the accuracy? This should be formalized.
Also, where does $t$ lie with respect to $i..j$ ?

### LACK OF CONTROL OF FPR IN EXPERIMENTS:
It is suggested that the lack of control of the FPR (which is often around 3% and sometimes above, while the specified level is 1%) is due to multiple hypothesis testing. However, I can't see which multiplicity you refer to. These rates are obtained sample-wise over the $R$ repetitions, and therefore they should be empirical estimates of the theoretical rate considered in Thm 3. A multiple hypothesis testing problem would occur if we use multiple statistics to get to one decision which is not the case here.
Three possible explanations come to my mind:
- a possible leak between the data used for training the Neural network and the data used to compute the statistic (i.e. some data points used to train a model $\hat{g}$ are also used as data points to compute the statistic with $\hat{g}$)
- the practical assumption that $\mathcal{S}$ is a uniform distribution over the training set : even if the training and test set come from the same distribution, this practical assumption will make $\mathcal{S}$ different from $\mathcal{T}$.
- A flaw in theorem 3?

Therefore I suggest to :
- Clarify the proof of Theorem 3
-  fix/give a better explanation of the uncontrolled FPR observed in the experiments
-  provide an empirical comparison to sequential two-sample tests: this would show the need for your approach and make the paper stronger

### MINOR ISSUES/COMMENTS:

1) Bayes Ball algorithm and reference to Bishop 2006:
Citation should be more precise since it is a text book . In fact, I couldn't find "Bayes Ball" in that book.
It seems to be called d-separation there.


2) "$\hat{g}\_t$ necessarily achieves a trivial accuracy of 1/2" : I guess you refer to theoretical accuracy. You should make it precise since there could a confusion with the achieved empirical accuracy which is not necessarily exactly 1/2.

3) The description of the algorithm should be improved: in particular, $w$ should be included as input and the formula for $\hat{\mu}_{w,t}$ should appear there.
The title says "Sequential Calibrated ...": the word "calibrated" appears nowhere else

4) THEOREM 5: it would be interesting to discuss the consequences of the assumptions $a(w,\alpha)\leq w$ and $b(w,\alpha)\geq 0$.

5) First inequality using the VC dimension: It would help the reader to give some justification/precise reference for this inequality.

6) Table 2b is misleading : higher FPRs are observed in the plots, when going beyond 200 samples

7) Proof of Thm 5: $\mu_{\hat{g}}^*$ seems to correspond to $\mu_{w,t}^*$ , why this new notation?

8) Equalities (8) and (9) should be inequalities.

9) Definitions should use a different equal sign ($\equiv$ or $:=$)


TYPOS:

"through online learned classifier" -> "through the online learned classifier"

covarate, covarite-> covariate

To checking -> for checking

Our estimate of the accuracy of $\hat{g}\_t$ **is** valid

Statement of theorem 5: "where $\epsilon\in(0,1/2]$ is the accuracy"  Maybe a comma is missing before "is"

Tables: G-shift -> GI-shift







**Summary Of The Paper:**

This paper proposes a new method to sequentially detect covariate shift which is the problem encountered when training and test covariates are not equally distributed. This methods builds on existing work that uses classifiers to distinguish between the two datasets and its accuracy as the statistic.
The contribution of this paper is proposing a sequential version, where the classifier is trained in an online manner  and the accuracy computed on a sliding window (of training and test samples) is used for each test set observation.
A criterion to reject the equality of the distributions in the sliding window is given using on the well known Clopper-Pearson interval. A lower bound on the probability of correctly accepting the null hypothesis is derived from the properties of the aforementioned interval. A lower bound on the probability of correctly rejecting is derived, under the assumption that the classifier has at least some discriminating power (i.e., that it is non trivial).


**Summary Of The Review:**

My main concern is the lack of control on the False Positive Rate (FPR) observed empirically. This is an important issue since it claims to have exact FPR control. This deserves further investigation since the given explanation based on multiple hypothesis testing is not convincing. The proof of the theoretical result that guarantees the FPR contains some obscure parts.
The paper also  lacks comparisons to existing sequential two-sample tests, which, even if  they do not provide guarantees on the False Negative Rate in non-iid scenarios,  are natural competitors since they do provide FPR guarantees (under the hypothesis of equal distributions).

Therefore, I recommend rejection in this current form.

**================== After rebuttal ==================**

The new comparison with Wald's test seems a bit limited by the alternative hypothesis with a fixed epsilon=0.2. Also, the power depends on the choice of the classifier. Cited nonparametric sequential two-sample tests are more flexible and should be considered instead.
My concerns on FPR control have been addressed.

For these reasons, I increase my score to 5.

---

> ### Author Response · Authors · 2021-11-23
> **Response to comments by Reviewer PsoX**
>
> Thank you for the valuable comments and concerns. The following includes our response; we have updated our paper accordingly and highlighted the updates in red.
>
> 1. COMPARISON TO SEQUENTIAL TWO-SAMPLE TESTS
> * **Response:** We have implemented the Sequential likelihood ratio test (Wald Test), and added the results to our paper. We described how we implemented the Wald test in Section 5.1, and included the testing result in Section 5.2 and 5.3. In summary, the test satisfies FPR bound, but is less effective at detecting shifts in all scenarios. Please see Figures 2 & 3 and Tables 2 & 3.
> We will also update the Wald test results for the additional experimental results in Appendix D as the experiments finish.
>
> 2. LEMMA 1:
> * **Response:** Here, $x_{i:j}$ means $x_i, …, x_j$, $\hat{g}\_i$  refers to a classifier itself (e.g., weights and biases of a neural network),
> and $ \hat{g}_{i:j-1}$ means the classifiers between time step $i$ and $j-1$. Also, $\hat{g}_i$ is a random variable since it depends on the previous inputs $x_i$ (as random variables) for training samples, but $\hat{g}_i$ itself has no randomness. We will more clearly describe this.
>
> 3. PROOF OF THEOREM 3
> * **Response:** $s_{w, t}$ is the sum of Bernoulli random variables and these random variables are independent due to the conditional independence of predictions $\hat{y}\_1, \dots \hat{y}\_t$ and $\hat{y} ’\_1, \dots \hat{y}’\_t$; thus $s_{w, t}$ is a Binomial random variable. Without this conditional independence we cannot say $s_{w, t}$ is a Binomial by the definition of the Binomial random variables (i.e., the sum of independent Bernoulli variables). For the “valid estimate of the accuracy”, we model the accuracy using the Binomial random variables, which is the true model, and estimating the accuracy is the same as the estimating the parameter of the Binomial distribution. Finally, $s_{w_t}$ will have parameter ½ only if there is no shift. We will clarify this.
>
> 4. LACK OF CONTROL OF FPR IN EXPERIMENTS:
> * **Response:** What we mean is that we are estimating FPR, so the FPR bound violations are the result of small experiment repetition (denoted by R), and this issue is exacerbated since we are evaluating FPR for many scenarios. If we used an infinite number of samples to estimate the FPR, then we would always meet the bound. In our previous version, we used R=1,000 repetitions to estimate FPR analysis, which resulted in violations. We have increased R to 20,000, and show the updated FPR results in Tables 2(b) and 3(b) in our paper (one table is for natural shift and the other one is for synthetic shift). As can be seen, with this higher choice of R, we have no FPR bound violations.
>
> MINOR ISSUES/COMMENTS:
>
> 5. Bayes Ball algorithm and reference to Bishop 2006: Citation should be more precise since it is a text book . In fact, I couldn't find "Bayes Ball" in that book. It seems to be called d-separation there.
> * **Response:** Thank you for pointing this out, we have fixed it.
>
> 6. "$\hat{g}_t$  necessarily achieves a trivial accuracy of 1/2" : I guess you refer to theoretical accuracy. You should make it precise since there could a confusion with the achieved empirical accuracy which is not necessarily exactly 1/2.
> * **Response:** Yes, this is theoretical accuracy. We will call it “expected accuracy”.
>
> 7. THEOREM 5: it would be interesting to discuss the consequences of the assumptions $a(w,α) \le w$ and $b(w,α) \ge 0$.
> * **Response:**  This condition is related to the valid range of w to be the bound meaningful (i.e., for the valid domain of the CDF F); we have updated the condition in a more concrete form. The condition means that the bound is valid if $w \ge 201$ given $\alpha = 0.01$. We will update the paper.
>
> 8. Table 2b is misleading : higher FPRs are observed in the plots, when going beyond 200 samples
> * **Response:** As pointed out, to compute the true FPRs we need an infinite number of random experiments; previously, we only used 1000 repetitions (i.e., $R=1000$), but we have increased the number of repetitions to 20000 (i.e., $R = 20000$); the updated Table 2b shows that our approach satisfies the desired FPRs empirically.
>
> 9. Proof of Thm 5: $\mu_{\hat{g}}^\*$  seems to correspond to $\mu\_{w,t}^\*$, why this new notation?
> * **Response:** This is a mistake. Thank you for pointing this out; we have fixed it.
>
> 10.  Definitions should use a different equal sign (≡ or :=)
> * **Response:** Thank you for pointing this out; we have fixed it.
>
> 11. Typos
> * **Response:** We appreciate pointing out these typos, and we fixed them.

---

> ### Author Response · Authors · 2021-12-02
> **Response to the After rebuttal**
>
> Thank you for your response updates. The followings are the answers to your additional comments.
>
> - We tried several values of $\epsilon$ (in particular, $\epsilon \in [0.01, 0.02, 0.05, 0.1, 0.15, 0.2, 0.25, 0.3, 0.35, 0.4]$). Our proposed approach clearly outperforms Wald tests with any of these choices of \epsilon (see here https://ibb.co/TM0FWck). We chose $\epsilon=0.2$ since it provided good results for both the "shift" and "no shift" segments of the back-and-forth scenario. Since Wald's test has memory, smaller values of $\epsilon$ tend to perform poorly when shifting from "shift" to "no shift". We will include results for all choices of $\epsilon$ in an update to our paper.
>
> - We have previously compared to non-classifier-based tests, which perform very poorly due to the high dimensionality of the data. For instance, in our updated paper, we show that our approach significantly outperforms the Deep Kernel Two-Sample Test, a state-of-the-art non-classifier-based two-sample test. We expect non-classifier-based sequential tests to suffer similar shortcomings, especially given that sequential tests are not suited to our setting.

---

### Official Review · Reviewer_yyGB · 2021-11-02

**Correctness:** 3
**Technical Novelty And Significance:** 2
**Empirical Novelty And Significance:** 3
**Recommendation:** 3
**Confidence:** 3

**Main Review:**

positives:
The authors investigate an essential problem in real-world applications. This draft is well-organized and easy to follow.

negatives:
- The proposed method attempts to adapt the offline covariate shift detecter to the sequential detection and learning scenario. However, due to the statistical nature of the two-sample test, the proposed approach still need a relatively large number of data to work. As shown in section 4, the error bound depends on the sample size in the order of $O(1/\sqrt{m})$. In the online learning scenario, the size of sampled (target) data is usually limited, and thus the effectiveness of this basic guarantee is limited.
- The experiments are conducted on the modified ImageNet dataset to simulate various covariate shifts. It is suggested to test the performance of the proposed algorithm on a realistic public benchmark dataset.

**Summary Of The Paper:**

This draft proposes an online covariate shift detection algorithm based on the two-sample test with source and target data in a fixed sliding window. The proposed method is based on the accuracy of the source-target classifier to identify the covariate shift: if the source-target classifier cannot distinguish the source or target data, then the proposed method considers they are generated from the same distribution and vice versa. Experiments on the modified ImageNet dataset validate the effectiveness of the proposed method.

**Summary Of The Review:**

This draft proposes an online covariate shift detection algorithm based on two-sample tests. The theoretical guarantee is based on the size of sampled data, and thus, it is usually hard to make the error (false positive/negative rate) small in real-world online applications. Empirical studies show good performance on the modified benchmark datasets with a relatively large number of sampled data.

---

> ### Author Response · Authors · 2021-11-23
> **Response to comments by Reviewer yyGB**
>
> Thank you for the valuable comments and concerns. The following includes our response; we have updated our paper accordingly and highlighted the updates in red.
>
> 1. The proposed method attempts to adapt the offline covariate shift detecter to the sequential detection and learning scenario. However, due to the statistical nature of the two-sample test, the proposed approach still need a relatively large number of data to work. As shown in section 4, the error bound depends on the sample size in the order of $O(1/\sqrt{m})$. In the online learning scenario, the size of sampled (target) data is usually limited, and thus the effectiveness of this basic guarantee is limited.
>
> * **Response:**  In general, $O(1/\sqrt{m})$ is the best sample complexity we can hope to achieve. Our experiments show that our algorithm is very effective in practice even with just a few examples: as demonstrated in Tables 2 (a) & 3 (a), our algorithm only requires a small number of examples to detect covariate shift in all scenarios.
>
>
> 2. The experiments are conducted on the modified ImageNet dataset to simulate various covariate shifts. It is suggested to test the performance of the proposed algorithm on a realistic public benchmark dataset.
>
> * **Response:** Section 5.2 is about a natural shift experiment which emulates a realistic shift scenario. In the experiment, we split 120 dog classes of ImageNet to two sets, and consider that the two subsets of the dog classes are source and target sets, respectively. In this setting, our approach is sample-efficient, and detects this shift well compared to the two baselines.

---

### Official Review · Reviewer_X4Zb · 2021-11-02

**Correctness:** 3
**Technical Novelty And Significance:** 3
**Empirical Novelty And Significance:** 3
**Recommendation:** 6
**Confidence:** 4

**Main Review:**

Pros:

1.  This research topic is valuable since the community cares about the AI reliability and trustworthy ML. Although deep networks can address many real-world problems, it is not reliable (e.g., OOD) and can be attacked (e.g., adversarial attacks). Thus, detecting such shifts is meaningful in many fields.

2. This paper proposed a test to dynamically detect the distribution shift, which seems novel and interesting.

3. Experiments are conducted in many situations, which supports the effectivenss of the proposed method.

Cons:

1. Some key discussions are missing. As reviewed in this paper, kernel MMD is another type of two-sample tests. If we use SGD or Adam to online update the kernel parameters, can kernel MMD be used to address your problem? It would be great to provide some empirical evidence, but detailed discussions are also ok.

2. In (1), what does \sP stand for?  It is a probability measure without proper explanations/definitions. For a probability measure P, we often use P(A) to represent the measure on a set A.

3. Some key references are missing. In the literature, concept drift detection and adaptation are very close to your problem setting. Please conclude some discussions between your problem and them.

4. How to generate the data used in this paper? What kind of shifts you considered? Mean shift? Covariance shift?


**Summary Of The Paper:**

In traditional machine learning, there is a basic assumption that training and test sets are from the same distribution. When this assumption holds, we can expect low prediction error in the test set. However, in the real world, this assumption may be broken. For example, when the images change from daylight to night, the distribution changes. As a result, we cannot fully trust a classifier trained with daylight images. To address this issue and make predictions reliable, detecting such covariate shifts seems promising. Although we can directly use a two-sample testing method to complete this detection task, it cannot fit the online requirement. This paper presents a new sequential classifier two-sample test to address these dynamic covariate shifts. This paper proves that their optimization preserves the correctness—i.e., the proposed algorithm achieves a desired bound on the false positive rate. In the experiments, they also show that the proposed algorithm efficiently detects covariate shifts on ImageNet.

**Summary Of The Review:**

In general, this paper considers an important problem. However, some points must be clarified before the possible acceptance.

---

> ### Author Response · Authors · 2021-11-23
> **Response to comments by Reviewer X4Zb**
>
> Thank you for the valuable comments and concerns. The following includes our response; we have updated our paper accordingly and highlighted the updates in red.
>
> 1. Some key discussions are missing. As reviewed in this paper, kernel MMD is another type of two-sample tests. If we use SGD or Adam to online update the kernel parameters, can kernel MMD be used to address your problem? It would be great to provide some empirical evidence, but detailed discussions are also ok.
>
> * **Response:** We have added experimental results to Appendix B; as can be seen, our approach significantly outperforms the adapted version of Kernel MMD. Due to time constraints (kernel MMD is slow), we only have results for Natural Shift; we will include complete results in the main paper once we have run them for all our scenarios.
>
> 2. In (1), what does \sP stand for? It is a probability measure without proper explanations/definitions. For a probability measure P, we often use P(A) to represent the measure on a set A.
>
> * **Response:** It is a probability measure on the event when the random variable s satisfies the specified condition. We will clarify this.
>
> 3. Some key references are missing. In the literature, concept drift detection and adaptation are very close to your problem setting. Please conclude some discussions between your problem and them.
>
> * **Response:** We thank you for suggesting additional references. Concept drift detectors typically assume that ground truth labels are provided for test examples, whereas our approach only requires unlabeled test examples. The former is substantially easier, since it suffices to check for drift in the distribution of errors made by the classifier, which is usually very simple (e.g., a Bernoulli distribution for the 0-1 loss), making it easy to exactly check for concept drift. In contrast, we are checking for drift in a high-dimensional covariate distribution. We have  added a discussion of concept drift detection in Section 2 on Related Work.
>
> 4. How to generate the data used in this paper? What kind of shifts you considered? Mean shift? Covariance shift?
>
> * **Response:** We are considering covariate shift (i.e., shift in p(x)). In our experiments, we consider two such shifts on ImageNet: natural shift and synthetic shift. For natural shift, we split 120 different dog breeds of ImageNet into two groups, and we consider the images from one subset of dog breeds to be the source distribution and ones from the other subset to be the target distribution. For synthetic shift, we have perturbed images by five different image transformations from ImageNet-C (Contrast, Defocus blur, Elastic transform, Gaussian blur, and Gaussian noise) with five different severities like Hendrycks and Dietterich (2019).
>
> Reference
> - Dan Hendrycks and Thomas Dietterich. Benchmarking neural network robustness to common corruptions and perturbations.Proceedings of the International Conference on Learning Representations, 2019.

---

> > ### Comment · Reviewer_X4Zb · 2021-12-05
> > **Thanks for the responses**
> >
> > Thanks for the responses. Most of my concerns are addressed well. Thus, I would like to increase my score from 5 to 6.
> >
> > However, I still have a concern regarding the type I error of DK. In DK, they use a permutation test to calculate the p-value. It should be around the alpha you chose. Could the authors explain this situation? It raises my concerns about if the generated data contain some dependency among data. If you set alpha to 0.05, the type I error of DK is still abnormal?

---

### Decision · Program_Chairs · 2022-01-20

**Decision:**

Reject

**Comment:**

This paper proposes to repeatedly apply the classifier two-sample tests (proposed by Kim, Ramdas, Singh, Wasserman, in 2016, and developed further by Lopez-Paz, Oquab, in 2017) for the purpose of detecting covariate shift. The authors propose methods to extend the aforementioned tests to a sequential setting. Overall, the reviewers do not lean towards acceptance, and neither do I. Several constructive suggestions are provided by reviewers, some are summarized below.

The authors claim that sequential tests are not desirable in such a setting, and thus choose to pay a multiple testing price by repeatedly applying a batch test. However, sequential tests are in fact applicable (they will control type-1 error) but may have a worse power if the alternative is not true at the very start --- but these were entirely dropped from the simulations; in fact, comparing the increased type-1 error of the authors' approach to the increased type-2 error of sequential approaches may be worth clarifying.

Perhaps the "right" solution that the authors are looking for could be gotten by converting a sequential test into a sequential changepoint detection algorithm (via repeated application of a sequential test, each started at a new time). Also see "Conformal test martingales for change-point detection" and "Inductive Conformal Martingales for Change-Point Detection" by Vovk et al., which are currently not cited.